# The structure of photosystem I from a high-light-tolerant cyanobacteria

Zachary Dobson[1,2], Safa Ahad[3], Jackson Vanlandingham[1,2], Hila Toporik[1,2], Natalie Vaughn[1,2], Michael Vaughn[1,2], Dewight Williams[4], Michael Reppert[3], Petra Fromme[1,2], Yuval Mazor[1,2]*

[1]School of Molecular Sciences, Arizona State University, Tempe, United States; [2]BiodesignCenter for Applied Structural Discovery, Arizona State University, Tempe, United States; [3]Department of Chemistry, Purdue University, West Lafayette, United States; [4]John M. Cowley Center for High Resolution Electron Microscopy, Arizona State University, Tempe, United States

**Abstract:** Photosynthetic organisms have adapted to survive a myriad of extreme environments from the earth's deserts to its poles, yet the proteins that carry out the light reactions of photosynthesis are highly conserved from the cyanobacteria to modern day crops. To investigate adaptations of the photosynthetic machinery in cyanobacteria to excessive light stress, we isolated a new strain of cyanobacteria, *Cyanobacterium aponinum* 0216, from the extreme light environment of the Sonoran Desert. Here we report the biochemical characterization and the 2.7 Å resolution structure of trimeric photosystem I from this high-light-tolerant cyanobacterium. The structure shows a new conformation of the PsaL C-terminus that supports trimer formation of cyanobacterial photosystem I. The spectroscopic analysis of this photosystem I revealed a decrease in far-red absorption, which is attributed to a decrease in the number of long- wavelength chlorophylls. Using these findings, we constructed two chimeric PSIs in *Synechocystis* sp. PCC 6803 demonstrating how unique structural features in photosynthetic complexes can change spectroscopic properties, allowing organisms to thrive under different environmental stresses.

**\*For correspondence:**
yuval.mazor@asu.edu

**Competing interest:** The authors declare that no competing interests exist.

## Introduction

Oxygenic photosynthesis evolved on earth about 2.5 billion years ago (*Bekker, 2004*). Plants, algae, and cyanobacteria carry out this process and are found in a wide variety of environments. Despite nearly 3 billion years of evolution, all oxygenic photosynthetic organisms use the same large pigment–protein complexes, known as photosystem I (PSI) and photosystem II (PSII), to convert solar energy to chemical energy (*Fromme et al., 2003*; *Witt, 1996*; *Zouni et al., 2000*). Both complexes use light to induce a charge separation event, then transport the high-energy electron, ultimately to be stored as a chemical bond (*Fromme et al., 2003*; *Witt, 1996*; *Zouni et al., 2000*). Although light is required for photosynthesis, an excess of light can be detrimental to photosynthetic organisms by damaging the photosynthetic proteins in a process called photoinhibition (*Kok, 1956*; *Jones and Kok, 1966*; *Gururani et al., 2015*). Therefore, the ability to adapt to different qualities and quantities of light is paramount for photosynthetic organisms to survive.

To prevent photoinhibition, photoprotective mechanisms such as changes in the photosystem content (*Muramatsu and Hihara, 2012*; *Murakami and Fujita, 1991*; *Demmig-Adams and Adams, 1992*; *Miskiewicz et al., 2002*), dissipation of excess energy as heat (*Peltier and Schmidt, 1991*; *Levy et al., 1993*; *Jacob and Lawlor, 1993*), and upregulation of antioxidant enzymes have been observed (*Tsang et al., 1991*; *Camp, 1994*; *Zhang and Scheller, 2004*; *Ort and Baker, 2002*; *Horling, 2003*). A response shown by the model cyanobacterium *Synechocystis* sp. PCC 6803 to high light is

to decrease the PSI:PSII ratio, minimizing its light harvesting capacity and presumably protecting against the generation of reactive oxygen species around PSI (*Hihara et al., 1998*; *Sonoike et al., 2001*). This is in agreement with the response observed in vascular plants (*Anderson, 1986*). However, this response is not universal. In the terrestrial cyanobacteria, *Synechococcus* OS-B′, isolated from a microbial mat in a hot spring, the ratio of PSI:PSII increases upon high-light conditions (*Kilian, 2007*), demonstrating the importance of PSI in the high-light response of cyanobacteria.

A significant amount of work on photoinhibition has focused on PSII due to its rapid turnover in high light and the efficient repair mechanisms that evolved to cope with PSII-specific photodamage (*Yao et al., 2012a*; *Yao et al., 2012b*). PSI-specific damage, however, is irreversible and long lived due to a lack of repair mechanisms, requiring the biosynthesis of new PSI polypeptides (*Li et al., 2004*; *Sonoike, 2010*; *Terashima et al., 1994*; *Sonoike et al., 1995*; *Lima-Melo et al., 2019*). Fluctuating light and low temperatures have been attributed to PSI photoinhibition by causing an imbalance in the redox state of PSI donors and acceptors (*Sonoike, 2010*; *Allahverdiyeva et al., 2015*; *Tiwari, 2016*; *Kudoh and Sonoike, 2002*; *Kono and Terashima, 2016*). Recovery times in leaves for PSI-specific photodamage have been reported to be longer than a week, much longer than the 30 min half-life of the D1 protein of PSII (*Zhang and Scheller, 2004*; *Yao et al., 2012b*; *Li et al., 2004*; *Kudoh and Sonoike, 2002*; *Zhou et al., 2004*; *Kanervo et al., 1993*). PSI inhibition is therefore potentially more devastating than damage to PSII because it results in the over-reduction of the plastoquinone pool subsequently inhibiting PSII and thereby blocking the complete electron transfer chain (*Sonoike, 2010*). This has lead to the proposal that photoinhibition of PSII is a mechanism to protect PSI by reducing the amount of electrons sent to PSI (*Barbato et al., 2020*; *Tikkanen et al., 2014*). Furthermore, there are multiple mechanisms acting on both the lumen and stromal side to reduce PSI damage under these conditions (*Sonoike, 2010*; *Tiwari, 2016*; *Barbato et al., 2020*; *Munekage, 2002*; *Munekage et al., 2008*; *Suorsa et al., 2013*; *Suorsa, 2015*), suggesting PSI damage plays an important role in adaptation to stress. The first high-resolution structure of cyanobacterial PSI revealed a core antenna system comprised of 96 chlorophyll *a* (Chl*a*) molecules, of which 6 are integral to the electron transport chain (ETC), while the remaining chlorophyll molecules function to harvest light and transfer excitation energy to the ETC (*Fromme et al., 2006*; *Jordan et al., 2001*). Still, the roles and properties of individual chlorophylls in the antenna remain largely unknown. While all chlorophyll molecules in cyanobacteria are chemically identical except one (one Chl*a* of P700 is Chl*a*′, the C13 epimer of Chl*a*), the local environment of a few individual chlorophylls has been shown to extend their absorbance properties above 700 nm, giving rise to long-wavelength chlorophylls (LWC) (*Fromme et al., 2006*; *Wientjes et al., 2012*; *Croce and van Amerongen, 2014*).

Further studies have revealed that, in cyanobacteria, LWC are associated with PSI and the amount of these pigments vary between species, suggesting that the number of LWC is an important evolutionary adaptation (*Shubin et al., 1991*; *Gobets et al., 2001*). It has also been shown that LWC are strongly affected by their immediate chemical environment. The absorbance properties of specific chlorophylls are highly dependent on its coordinating residues as well as excitonic coupling between neighboring chlorophylls; and changing either can alter the spectroscopic properties of a chlorophyll. (*Wientjes et al., 2011*; *Toporik, 2020*; *Khmelnitskiy et al., 2020*). These characteristics have led to several suggestions to the physiological role of LWC such as directing energy to P700, extending light-harvesting capabilities into the far red, and photoprotective mechanisms (*Valkunas et al., 1995*; *Trissl, 1993*; *Rivadossi et al., 1999*; *Schlodder et al., 2005*; *Herascu et al., 2016*).

While the function of PSI is conserved in all photosynthetic organisms, the oligomeric state and subunit composition of PSI varies. Higher plants exclusively utilize monomeric PSI, whereas cyanobacteria utilize monomeric, trimeric, and tetrameric oligomers (*Li et al., 2014*; *Li et al., 2019*; *Zheng et al., 2019*). It has been suggested that trimerization is a way of modulating light harvesting in changing light conditions (*Chitnis and Chitnis, 1993*; *Sener et al., 2004*). The PsaL subunit has been shown to be vital for these larger oligomeric complexes to form (*Li et al., 2019*; *Zheng et al., 2019*; *Chitnis and Chitnis, 1993*; *Malavath et al., 2018*). Adding a single histidine to the C-terminus of PsaL was shown to completely disrupt trimerization in Synechocystis (*Malavath et al., 2018*; *Netzer-El et al., 2018*). The first crystal structures of PSI from *Thermosynechococcus elongatus* and *Synechocystis* sp. PCC. 6,803 (*Synechocystis*) showed that the C-terminus of PsaL coordinates a calcium ion together with the PsaL subunit of the adjacent monomer, which was suggested to stabilize trimer

formation and emphasized the importance of the C-terminus of PsaL in oligomerization (*Jordan et al., 2001*; *Mazor et al., 2013*).

In order to understand how the light harvesting machinery has evolved to adapt to high-light conditions, we isolated a cyanobacterium from the Sonoran Desert, an environment with light intensities regularly exceeding 1600 µmol photon m$^{-2}$s$^{-1}$, in order to characterize its photosynthetic machinery. Genomic sequencing revealed that this cyanobacterium is a new strain of *Cyanobacterium aponinum*. Other strains of *C. aponinum* have been shown to grow in both freshwater and seawater, as well as extreme environments with temperatures reaching 45°C (*Moro et al., 2007*; *Winckelmann et al., 2015*). The ability of *C. aponinum* to survive in these vastly different environments make it a promising candidate for biofuel production (*Ertugrul and Dönmez, 2011*). Here we report the structure and the spectroscopic characterization of the trimeric PSI complex isolated from the high-light tolerant *C. aponinum*. We designed two chimeric PSIs in *Synechocystis* to test the functionality of structural variations between *C. aponinum* and *Synechocystis*. Our results demonstrate how the structure of PSI modulates its spectroscopic properties and elucidate the mechanisms controlling PSI oligomerization in cyanobacteria, bringing the ability to design large photosynthetic complexes with desired optical properties a step closer.

## Results

### *C. aponinum* 0216 is a high-light-tolerant cyanobacteria

To study a photosynthetic organism that exhibits the ability to grow in high-light environments, samples were taken from a biofilm growing on a south facing concrete wall of a freshwater reservoir in Tempe, AZ, that had a constant drip of fresh water and exposed to over 300 days of sunlight per year. Samples were taken in February, which has an average temperature of 18.7 °C according to the National Climatic Data Center (NOAA) for this area (*LOCAL CLIMATOLOGICAL DATA, 1946*). Samples were cultivated in BG-11 growth media and exposed to light intensities exceeding 3000 µmol photon m$^{-2}$s$^{-1}$ of warm white light at 30 °C for a week to select for organisms able to survive high-light conditions. One photoautotroph was able to survive these light intensities and was subsequently isolated through continuous streaking on BG-11 media agar plates supplemented with iron (*Figure 1*). Genomic DNA was extracted, and the 16 S rRNA was amplified to identify the organism (*Nubel et al., 1997*). The 16 S rRNA was compared to other cyanobacterial 16 s rRNA libraries revealing a close relationship to *C. aponinum* strains (*Figure 1B*).

The ability of *C. aponinum* to grow in high light was compared directly to *Synechocystis* across several light intensities (*Figure 1C*). Cells were serially diluted and exposed to a range of light intensities to determine the viability for growth. *C. aponinum* grew in much higher light intensities than *Synechocystis*, with the ability to survive in conditions as high as 1850 µmol photons m$^{-2}$s$^{-1}$. Under these conditions, *Synechocystis* cannot grow regardless of the density of the cells on the culture plate, thus emphasizing the innate ability of *C. aponinum* to grow under high-light conditions.

The response of *C. aponinum* to high-light conditions was measured by comparing *C. aponinum* grown under high- (450 µmol photons m$^{-2}$s$^{-1}$) and low-light (45 µmol photons m$^{-2}$s$^{-1}$) conditions. Absorption spectra revealed that cells grown in high light show an increase in absorption between 400 and 550 indicating a higher carotenoid content relative to chlorophyll, which is a known response to high light in photosynthetic organisms (*Sozer, 2010*; *Figure 1D*). Additionally, low-temperature fluorescence measurements (77 K) indicated that the F722:F685 ratio, a proxy for the distribution of excitation energy between PSI:PSII in vivo (*Murakami, 1997*), increases from 2.00 in low-light cells to 3.14 in cells grown in high light (*Figure 1E*). This increase under high light led us to investigate PSI and determine if it is involved in the high-light tolerance observed in *C. aponinum*.

### C. *aponinum* PSI contains less LWC and has a modified PsaL

To explore possible adaptations of PSI to high light, the complex was isolated using anion exchange chromatography of solubilized thylakoid membranes. The sucrose density gradient shows the chlorophyll-containing species (*Figure 2A*). Comparing this sample to a known PSI sample from *Synechocystis*, SDS–PAGE shows similar bands for PSI subunits in *Figure 2B*, with notable shifts in the PsaD, PsaF, PsaL, and PsaC subunits.

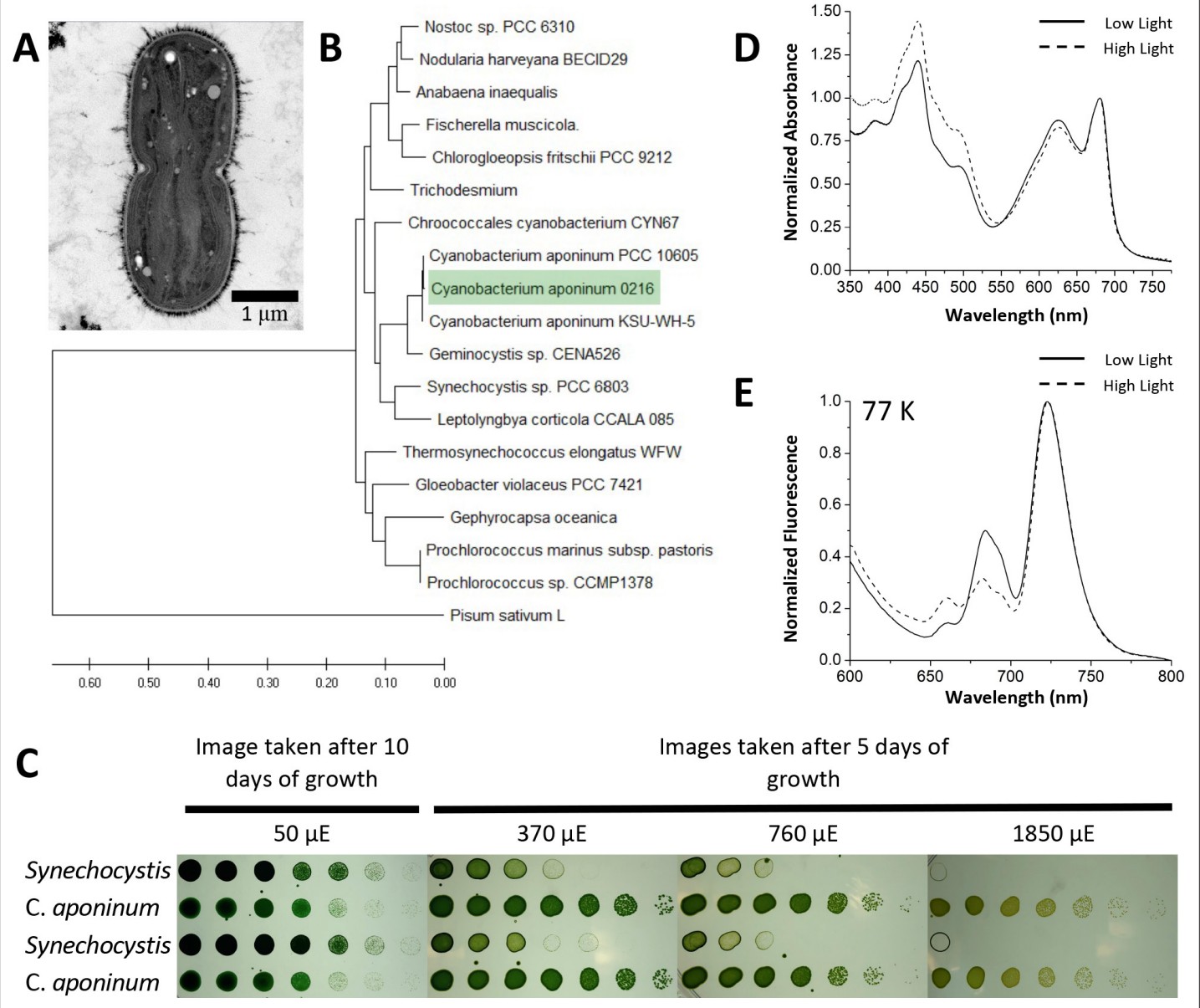

**Figure 1.** Isolation of a high-light-tolerant cyanobacteria. (**A**) Cross-sectional negative stained image of *C. aponinum* fixed in acrylic medium. (**B**) Phylogenetic analysis based on *C. aponinum* 16 S rRNA. Evolutionary analyses were conducted in MEGA7. (**C**) Serial dilutions of *Synechocystis* and *C. aponinum* on BG11 plates. Cells were serially diluted in ¼ steps and incubated at 30 °C for 5 days (light intensities > 370 µmol photons m⁻²s⁻¹) and 10 days (light intensity = 50 µmol photons m⁻²s⁻¹) (**D**) in vivo absorption spectra (normalized to the max wavelength of the Q$_y$ transition) of *C. aponinum* cells grown in low light (45 µmol photons m⁻²s⁻¹) and high light (450 µmol photons m⁻²s⁻¹). (**E**) 77 K fluorescence spectra (normalized to the max emission wavelength) of whole cells excited at 440 nm.

The online version of this article includes the following source data for figure 1:

**Source data 1.** Source data for *Figure 1*.

To investigate the different migration of PSI subunits between *C. aponinum* and *Synechocystis*, *C. aponinum* genomic DNA was isolated and sequenced (NCBI:txid2676140). PSI genes were located, annotated, and compared to *Synechocystis* (*Figure 2—figure supplement 1*). The sequence of the PsaL gene revealed that the difference in migration is likely due to two substantial differences compared to *Synechocystis*: (1) a six amino acid insert located on the stromal side of the membrane between two transmembrane helixes and (2) a markedly different C-terminus (*Figure 2—figure supplement 1*) containing an extension of four amino acids. In addition,

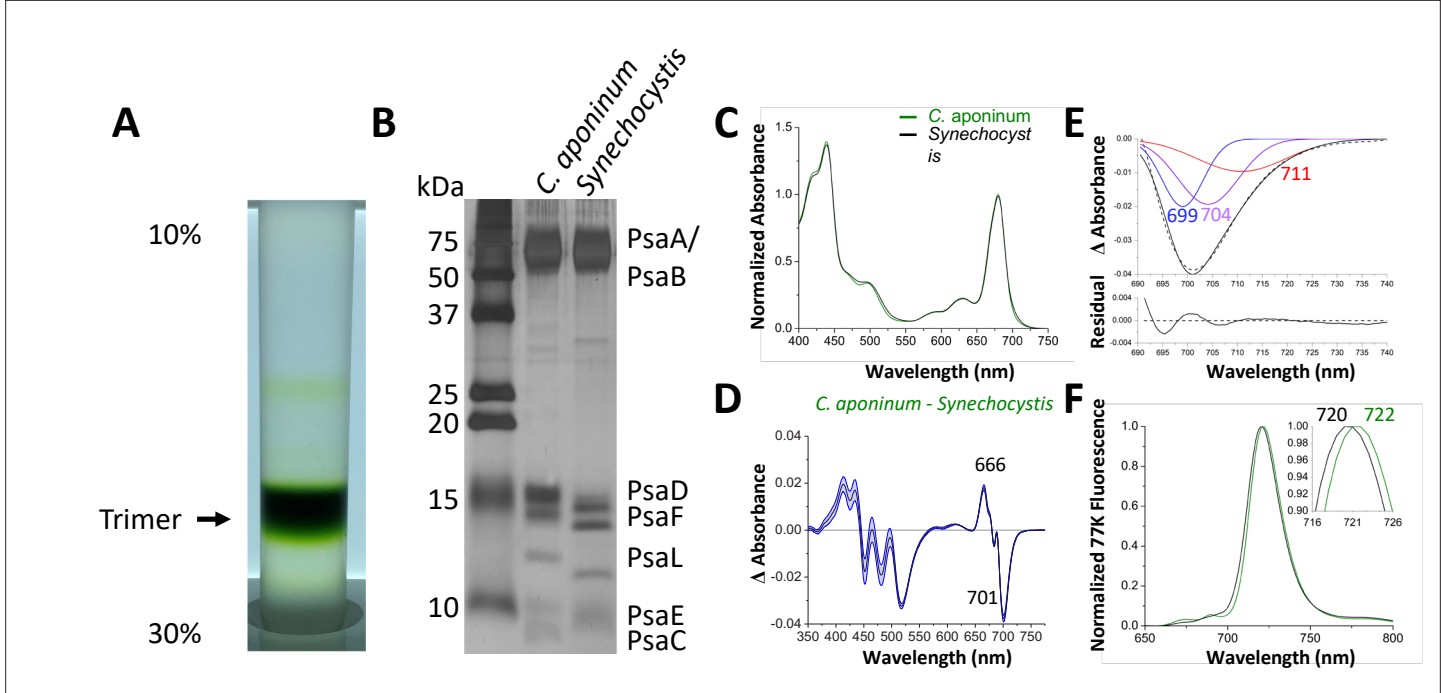

**Figure 2.** Isolation and characterization of trimeric PSI. (**A**) Ten percent to 30% sucrose gradient of solubilized *C. aponinum* membranes following an anion exchange chromatography. (**B**) SDS–PAGE of the main sucrose gradient band (Trimer) compared to PSI isolated from *Synechocystis*. A notable difference between the PsaL bands is clearly observable around 13 kD. (**C**) Absorption spectra of the purified trimer of *C. aponinum* (green) and *Synechocystis* (black) normalized to the area between 550 and 775 nm (**D**) Difference of the *C. aponinum – Synechocystis* absorbance spectra, shown are averages of three biological replicas, blue edges indicate± SD. (**E**) The negative peak at 701 nm of the absorbance difference spectrum (dashed black line) is fitted to a sum of three gaussian components colored blue, purple, and red with the sum as a solid black line and the residual of the fit (**F**) 77 K fluorescence of *C. aponinum* (green) and *Synechocystis* (black) using an excitation wavelength of 440 nm. Samples were normalized to their max peak.

The online version of this article includes the following source data and figure supplement(s) for figure 2:

**Source data 1.** Source data for *Figure 2*.

**Figure supplement 1.** Photosystem I protein sequence alignment comparison between *C.*

a seven amino acid insertion is seen in the PsaB gene of *C. aponinum* compared to *Synechocystis* (*Figure 2—figure supplement 1*). Genes for the remaining subunits (PsaD, PsaF, and PsaC) did not reveal differences that would correspond to these shifts (*Figure 2—figure supplement 1*). There were however sequence variations between *C. aponinum* and *Synechocystis*, which could cause gel shifting, a common occurrence when analyzing membrane proteins via SDS–PAGE (*Rath et al., 2009*).

Absorption spectra from PSI purified from *C. aponinum* and *Synechocystis* were normalized to the area between 550 nm to 775 nm to account for the individual contribution of each chlorophyll (*Figure 2C*). The difference spectrum of the absorption, *C. aponinum–Synechocystis*, shows a strong negative peak at 701 nm revealing that *C. aponinum* PSI contains less LWC than *Synechocystis* (*Figure 2D*). Based on the Gaussian deconvolution of this peak three components were identified at 699, 704, and 711 nm, the latter two in agreement with the site energy of previously reported LWC in *Synechocystis* that would be altered in *C. aponinum* (*Toporik, 2020*; *Khmelnitskiy et al., 2020*; *Figure 2E*).

Surprisingly, the 77 K emission spectra shows that the emission peak of *C. aponinum* PSI displays a 2 nm red shift compared to *Synechocystis* PSI (*Figure 2F*). We attribute this shift to enhanced emission from a red state common to both complexes, as we only detected loss of LWC in *C. aponinum* compared to *Synechocystis*. To identify which chlorophylls could be responsible for the different spectroscopic properties, we determine the structure of the PSI trimer from *C. aponinum* using Cryo-EM.

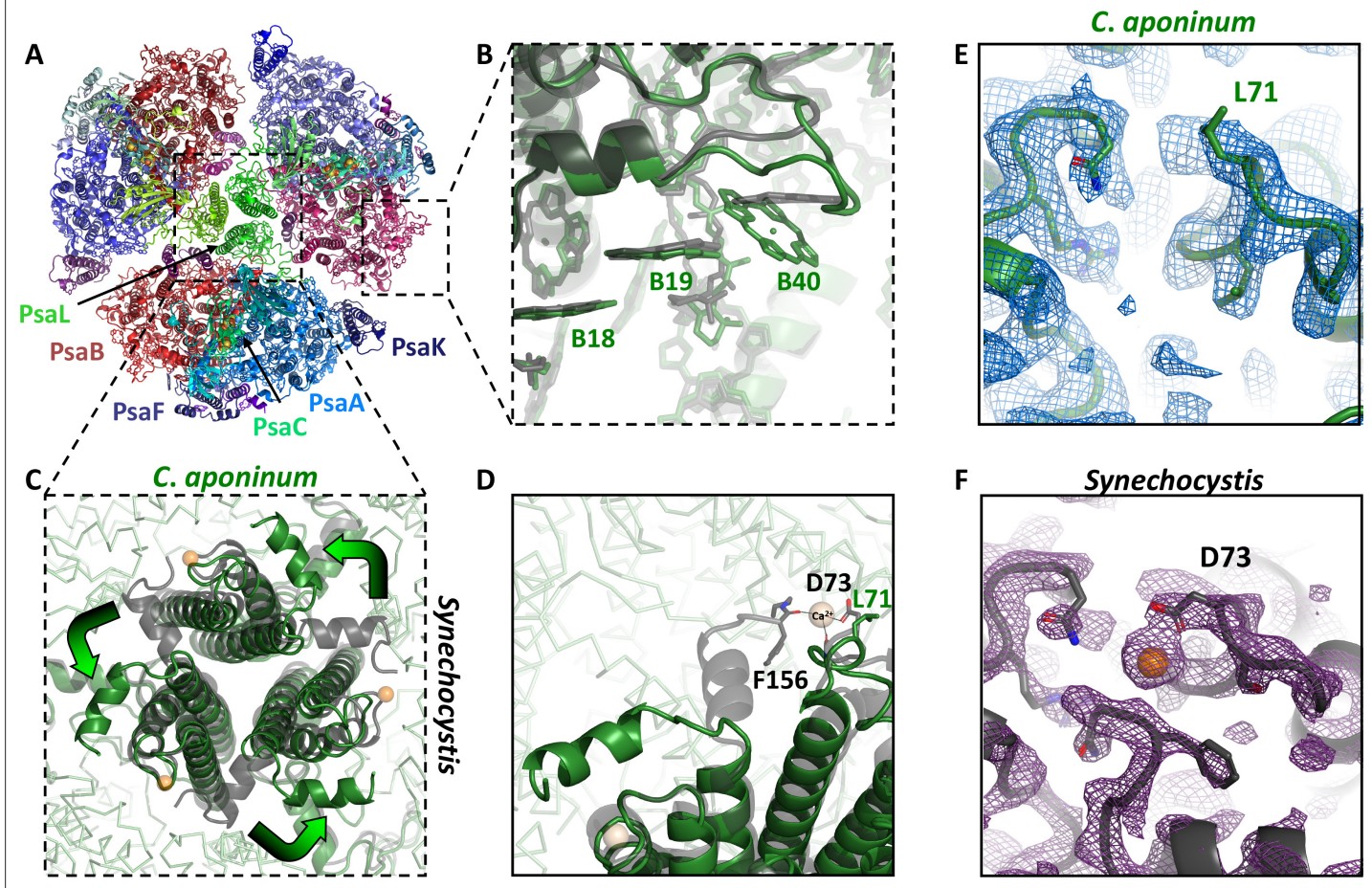

**Figure 3.** The structure of trimeric PSI from *C.* aponinum. (**A**) *C. aponinum* trimeric PSI (**B**) chlorophyll B40 shifts its position due to the insertion seen in the PsaB subunit in *C. aponinum* (green) compared to *Synechocystis* (black). (**C**) The PsaL subunits of *C. aponinum* (green) and *Synechocystis* (black) showing the difference of the overall structure of the PsaL C-terminus. (**D**) The C-terminus of the PsaL subunit of *C. aponinum* (green) and *Synechocystis* (black) displaying the coordination to the $Ca^{2+}$ in the adjacent monomer in *Synechocystis*, but is absent in *C. aponinum* and the *Red_d* mutant of *Synechocystis*. (**E**) *C. aponinum* and its electron density map compared to (**F**) *Synechocystis* (PDBID 5OY0, shown with 2Fo-Fc map) clearly depicting no density for the $Ca^{2+}$ ion in the map for *C. aponinum*.

The online version of this article includes the following source data and figure supplement(s) for figure 3:

**Source data 1.** Source data for *Figure 3*.

**Figure supplement 1.** Cryo-EM data processing.

**Figure supplement 2.** Model resolution and map examples.

**Figure supplement 3.** The local environment of Chl1240.

**Figure supplement 4.** Comparison of the PsaL subunits from *C.*

**Figure supplement 5.** $Ca^{2+}$ coordinating residue in prokaryotes.

**Figure supplement 6.** A bar graph representing the frequency of different amino acids in eukaryotes, and their side chain properties, at the position which would coordinate the $Ca^{2+}$ ion observed in early cyanobacteria structures.

## Structure of trimeric PSI from *C. aponinum*

Using single-particle cryo-EM, the structure of trimeric *C. aponinum* PSI seen in *Figure 3A* was determined to 2.7 Å resolution as assessed by the gold-standard Fourier shell correlation (FSC) criteria (*Henderson, 2011*) when imposing C3 symmetry (*Table 1* and *Figure 3—figure supplements 1 and 2*). Local resolution calculations show that regions of the interior of the protein are resolved to the Nyquist limit of 2.1 Å, while surface- and membrane-exposed regions are resolved to 2.5 Å resolution (*Figure 3—figure supplement 2*). The complex contains 33 protein subunits, 288 chlorophylls, 72 carotenoids, 6 phylloquinones, and 9 iron–sulfur clusters as shown in *Figure 3A*. Interestingly, no

**Table 1.** Cryo-EM data collection, refinement, and validation statistics.

| | PSI complex (EMD-21320, PDB-6VPV) |
|---|---|
| **Data collection and processing** | |
| Calibrated pixel size (Å) Detector, physical pixel size (µm) | 1.05 K2 summit, 5 |
| Voltage (kV) | 300 |
| Total electron dose (e−/Å²) | 61 |
| Defocus range (µm) | −1.5 to − 3.0 |
| Super pixel size (Å) | 0.525 |
| Symmetry imposed | C3 |
| Initial particle images (no.) | 256,410 |
| Final particle images (no.) | 73,984 |
| Map resolution (Å) | 2.7 |
| FSC threshold | 0.143 |
| Map resolution range (Å) | 2.1–4.1 |
| **Refinement** | |
| Initial model used (PDB code) | 5OY0 |
| Model resolution (Å) | 2.7 |
| FSC threshold | 0.143 |
| Model resolution range (Å) | 2.1–4.1 |
| Map sharpening B factor (Å²) | −72.48 |
| Model composition | |
| Nonhydrogen atoms | 71,814 |
| Protein residues | 6,743 |
| Ligands | 384 |
| B factors (Å²) | |
| Protein | 50.00/137.33/86.61 |
| Ligand | 27.10/131.48/54.64 |
| R.m.s. deviations | |
| Bond lengths (Å) | 0.005 |
| Bond angles (°) | 0.894 |
| **Validation** | |
| MolProbity score | 1.82 |
| Clashscore | 10.36 |
| Poor rotamers (%) | 0.0 |
| Ramachandran plot | |
| Favored (%) | 95.89 |
| Allowed (%) | 4.11 |

*Table 1 continued on next page*

*Table 1 continued*

| | PSI complex (EMD-21320, PDB-6VPV) |
|---|---|
| Disallowed (%) | 0 |

density was observed for the $Ca^{2+}$ ion in the PsaL subunit that was proposed to stabilize trimerization in previously solved trimeric PSI structures (*Figure 3E,F*; *Jordan et al., 2001*; *Mazor et al., 2013*).

Comparison of the PSI chlorophyll arrangement between *C. aponinum* and *Synechocystis* reveals a high degree of conservation, except for chlorophyll B40 (*Figure 3B*). This chlorophyll is located next to a seven amino acid insertion seen in the sequence alignment of PsaB (*Figure 3—figure supplement 3*) that creates a loop sterically forcing chlorophyll B40 to shift its orientation in *C. aponinum*. As a result, the coupling between chlorophyll B40 and B19 changes. The significance of this conformation was determined using a combination of mutagenesis and modeling (see below). Shorter insertions at the same PsaB location are also observed in PSI sequences from other photoautotrophs, including *T. elongatus* and *Pisum sativum*, marking this PsaB loop as a unique, variable region, in the core PSI (shown in *Figure 3—figure supplement 3*). The structure of *T. elongatus* PSI shows the corresponding B40 chlorophyll is not present due to an additional subunit, PsaX, which results in a different conformation of this loop relative to *C. aponinum* (*Jordan et al., 2001*), sterically blocking the binding site of chlorophyll B40. In the structure of *P. sativum* PSI, chlorophyll B40 is present; however, it exhibits a similar shift compared to *C. aponinum* because of a two amino acids insertion (*Figure 3—figure supplement 3*).

In addition to the rearrangement around chlorophyll B40, an additional chlorophyll molecule was modeled on the stromal side of the PsaK subunit in the *C. aponinum* structure. This chlorophyll was not modeled in early cyanobacteria PSI structures; however, it has been recently resolved in PSI structures from *Synechocystis* and *Synechococcus* sp. PCC 7942 (henceforth *Synechococcus*) (*Toporik, 2020*; *Cao et al., 2020*).

The PsaL subunit in *C. aponinum* reveals two drastically different features compared to *Synechocystis*: First, a large loop on the stromal side of the membrane (*Figure 3—figure supplement 4*) and second, a C-terminus that does not bridge adjacent monomers through the coordination of a $Ca^{2+}$ ion (*Figure 3C,D*; *Chitnis and*

*Chitnis, 1993*; *Malavath et al., 2018*). The loop on the stromal side in *C. aponinum* lays relatively flat along the membrane plane and only differs slightly in conformation compared to plant PSI due to the PsaH subunit, which is missing in cyanobacteria. In the plant PSI structure this loop is raised to accommodate the binding of PsaH shown in *Figure 3—figure supplement 4*.

Previously solved PSI structures from cyanobacteria reveal that the C-terminus of PsaL from one monomer coordinates a $Ca^{2+}$ ion together with a negatively charged residue from the adjacent PsaL subunit (*Jordan et al., 2001*; *Mazor et al., 2013*). However, in the structure from *C. aponinum* this interaction does not occur, as shown in *Figure 3C*. Comparing the structures of *Synechocystis* and *C. aponinum* revealed that at the position of aspartate 73, one of the $Ca^{2+}$ coordinating residues in *Synechocystis*, a leucine is present in *C. aponinum* (*Figure 3D*). Unlike aspartate, leucine is an uncharged species which prevents the coordination of this $Ca^{2+}$ ion. Furthermore, the C-terminus in *C. aponinum* is longer than *Synechocystis* (*Figure 2—figure supplement 1* and *Figure 3—figure supplement 4*) and would sterically prevent oligomerization if adopting the same orientation as *Synechocystis*.

A similar orientation of PsaL was shown in the structure of the PSI-IsiA antenna super-complex from *Synechococcus*. In this structure, there is also no $Ca^{2+}$ ion modeled and an asparagine is in the coordination position (*Cao et al., 2020*). To understand the frequency of cyanobacterial PSI that coordinate a calcium ion, a protein alignment of 680 cyanobacteria PsaL sequences was constructed and sorted by their position on an evolutionary tree. This alignment revealed that the position of the calcium coordinating residue is variable in cyanobacteria. Early structures suggest a negatively charged residue is crucial for the calcium coordination (*Jordan et al., 2001*; *Malavath et al., 2018*); however, negatively charged residues were present in only 48 % of sequences in this alignment (*Figure 3—figure supplement 5*). The next most prevalent residue in this position is asparagine, occurring in 30 % of the sequences. The recent PSI-IsiA structure from *Cao et al., 2020* contains an asparagine at this position, demonstrating that asparagine does not coordinate an calcium ion. Additionally, in eukaryotic organisms, which contain monomeric PSI, the residue at this position is as asparagine in 81 % of sequences (*Figure 3—figure supplement 6*). Therefore, other interactions must be present for trimerization to occur, as less than 50 % of cyanobacteria contain a residue capable of coordinating $Ca^{2+}$ at this position.

## Structural changes lead to spectral shifts around the $Q_y$ transition

To explore the functional significance of the structural differences highlighted in *Figure 3* between *C. aponinum* and *Synechocystis* (henceforth *WT Synechocystis*), two mutant strains of *Synechocystis* were constructed. One, *Red_c*, contains the sequence of the PsaB loop from *C. aponinum* (*Figure 3B*, *Figure 3—figure supplement 3*) and the other, *Red_d*, contains a point mutation at the calcium coordinating aspartic acid in PsaL to a leucine as observed in *C. aponinum* (*Figure 3D–F*).

Trimeric PSI was purified from both *Red_c* and *Red_d* (*Figure 4A*). Subsequence SDS–PAGE shows that these samples have the same subunit composition seen in the *WT Synechocystis* PSI (*Figure 4B*). Absorbance spectra were normalized to the area between 550 nm and 775 nm (*Figure 4C*), and the spectrum of *WT Synechocystis* PSI was subtracted from both *Red_c* and *Red_d* (*Figure 4D*).

The difference spectrum of *Red_c – WT Synechocystis* around the chlorophyll $Q_y$ transitions revealed a positive peak with a maximum at 669 nm and a negative peak with a minimum at 685 nm, but relatively little change in wavelengths above 700 nm. We attribute these differences to the altered orientation of chlorophyll B40 in *Red_c (see below)*. The difference spectrum of *Red_d – WT Synechocystis* (*Figure 4D*) showed that removal of the $Ca^{2+}$ clearly affect the absorption of PSI in the far-red region of the Qy transition, evident by the negative peak centered at 704 nm. In contrast to affecting individual chlorophyll coordination or orientation (by point mutations or loop insertion) removing the positive charge of the $Ca^{2+}$ ion is expected to have a more global effect, possibly influencing several chlorophylls in different ways (depending on or their orientation and distance from the $Ca^{2+}$ ion). We attribute the additional features in the *Red_d* difference spectra to interactions between the $Ca^{2+}$ and neighboring chlorophylls which are not part of the LWC in PSI. The identity of the LWC affected by the removal of $Ca^{2+}$ ion is discussed further below.

Further confirmation that LWC absorption is modified by $Ca^{2+}$ binding in PSI is seen in room temperature emission from PSI. While the emission from trimers isolated from *Red_c* strains is similar to the wild-type trimer at the red regions of the spectra, the emission from PSI isolated from *Red_d*

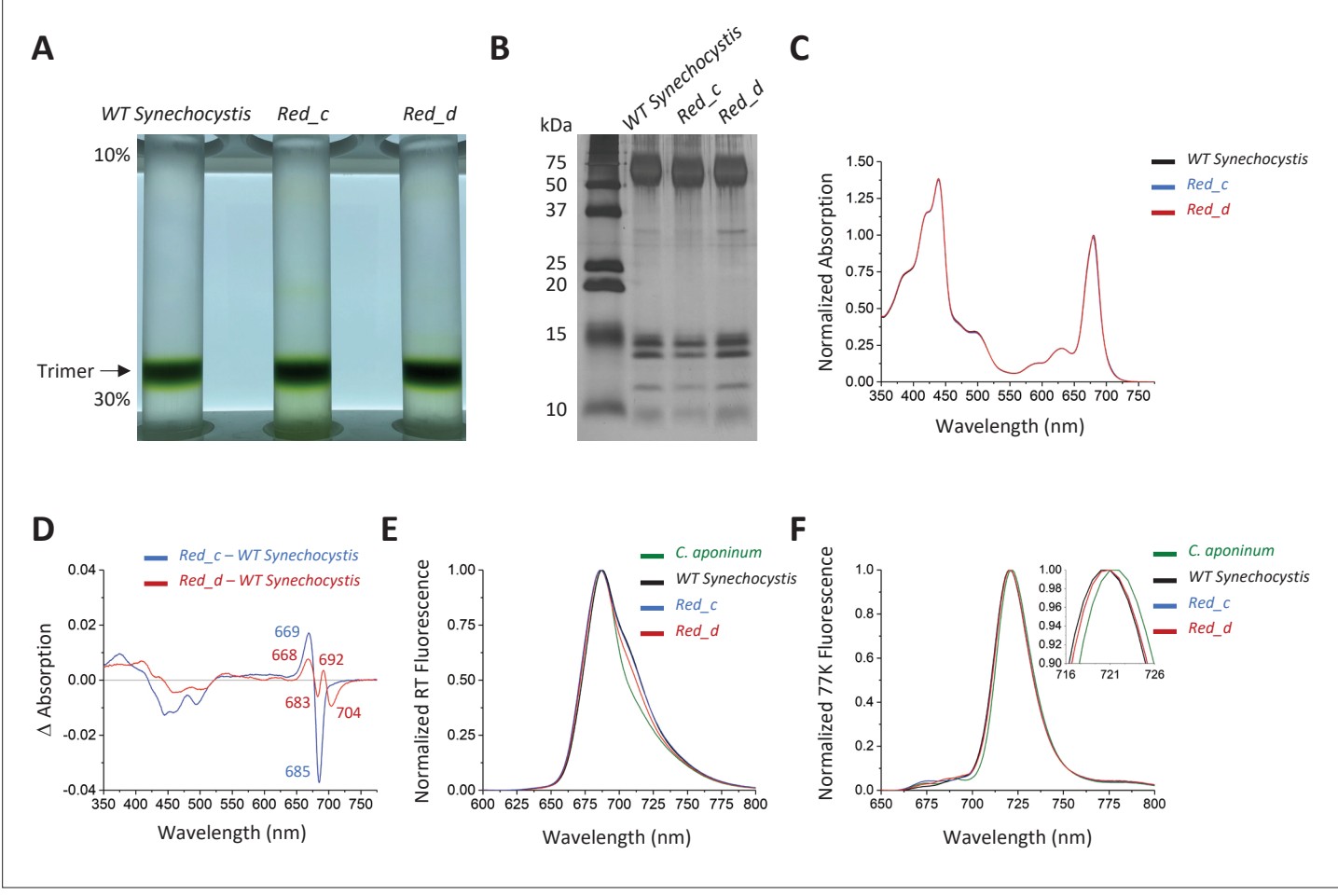

**Figure 4.** Spectroscopic analysis of *Red_c*. (**A**) Ten percent to 30% sucrose gradient of solubilized membranes from *WT Synechocystis*, *Red_c,* and *Red_d* after purification by anion exchange. (**B**) SDS–PAGE of the main sucrose gradient bands in comparison with PSI isolated from *WT Synechocystis*. (**C**) Absorption spectra of the purified trimer of *Red_c mutant* (blue), *Red_d* (red), and *WT Synechocystis* (black) normalized to the area between 550 and 775 nm (**D**) Difference spectra of the *Red_c – WT Synechocystis* (blue) and the *Red_d – WT Synechocystis* absorbance spectra (red). (**E**) Room temperature emission using an excitation wavelength of 440 nm. Samples were normalized to their max peak. (**F**) 77 K fluorescence of *C. aponinum* (green), *WT Synechocystis* (black), *Red_c* (blue), and *Red_d* (red) using an excitation wavelength of 440 nm. Samples were normalized to their max peak.

The online version of this article includes the following source data and figure supplement(s) for figure 4:

**Source data 1.** Source data for *Figure 4*.

**Figure supplement 1.** Solubilized membranes of *WT Synechocystis* and *Red_d* before anion exchange, displaying the increased relative ratio of PSI monomers to trimers in vivo in *Red_d*.

strains is different and its intensity at the red region of the spectra is significantly lower than emission from wild-type trimers (*Figure 4E*). Surprisingly, despite removing LWC in the *Red_d* strain, we did not resolve differences in the 77 K emission peak between *WT Synechocystis* and *Red_d* (*Figure 4F*). These remaining differences show that additional mutations are needed to fully account for the differences between the two strains (see Discussion).

It was previously shown that disruption (deletion or C terminal extension) of the PsaL subunit prevents trimerization in PSI (*Chitnis and Chitnis, 1993*; *Malavath et al., 2018*). A PSI trimer was readily isolated from *Red_d*, showing that $Ca^{2+}$ binding is not essential for trimer formation. However, we investigated the distribution of the trimer configuration in native membranes by solubilizing and running a sucrose gradient without additional purification steps (*Figure 4—figure supplement 1*). These experiments showed a higher monomer to trimer ratio in the *Red_d* strain compared to *WT Synechocystis*, suggesting that although $Ca^{2+}$ is not required for trimerization, $Ca^{2+}$ coordination does play a role in stabilizing the trimeric organization in *WT Synechocystis*.

**Table 2.** Site energies and calculated coupling values amongst chlorophylls B18, B19, and B40 for *WT Synechocystis* and *C. aponinum* structures.

Diagonal entries (in bold) are site energies used in excitonic structure calculations. Red entries above the diagonal are calculated from the *WT Synechocystis*; blue entries below the diagonal are for *C. aponinum*. All values are in cm$^{-1}$.

|  | *B18* | *B19* | *B40* |
|---|---|---|---|
| *B18* | **14,600** | −71 | −20 |
| *B19* | −59 | **14,950** | −106 |
| *B40* | −10 | −70 | **14,950** |

## Modified excitonic interactions explains the observed spectral differences

The shape of the difference spectra between *Red_c* and *WT Synechocystis* (*Figure 4D*) is easily understood as a consequence of oscillator strength redistribution amongst a coupled cluster of chlorophyll pigments, an effect often observed in photosynthetic hole burning spectra (*Reppert et al., 2010*; *Reppert et al., 2009*; *Reppert et al., 2008*). Briefly, the shift in conformation of chlorophyll B40 is expected to modify both its transition dipole orientation and excitonic coupling interactions with its neighboring pigments, particularly chlorophylls B18 and B19. These altered interactions affect both the transition energies and oscillator strengths of the B18/B19/B40-cluster exciton states, presumably (based on the experimental absorption difference spectrum) shifting absorption intensity from a low-energy exciton near 686 nm to a higher-energy band near 670 nm (*Reppert et al., 2010*; *Reppert et al., 2009*; *Reppert et al., 2008*).

To test this explanation, we calculated electronic transition dipoles and coupling elements amongst chlorophylls B18, B19, and B40 using the transition electrostatic potential (TrESP) method (*Madjet et al., 2006*). For the TrESP calculation, we used the gas-phase transition charges previously calculated (*Madjet et al., 2006*) and rescaled here to ensure that each pigment carried a Q$_y$ dipole moment strength of 4.3 Debye (*Reppert et al., 2010*; *Reppert et al., 2008*). Calculated coupling values are displayed in *Table 2* and confirm that excitonic interactions are significantly modified between the two structures. In the table, entries above the diagonal correspond to coupling elements in *WT Synechocystis*, and values below the diagonal correspond to the *C. aponinum* structure, presumed to be similar to the *Red_c* mutant; all values are in units of cm$^{-1}$. The largest difference is the B19/B40 coupling element which decreases in magnitude from −106 cm$^{-1}$ in the WT structure to only −70 cm$^{-1}$ in the *Red_c* mutant. In addition, as illustrated in *Figure 5A,B*, the transition dipole moment for

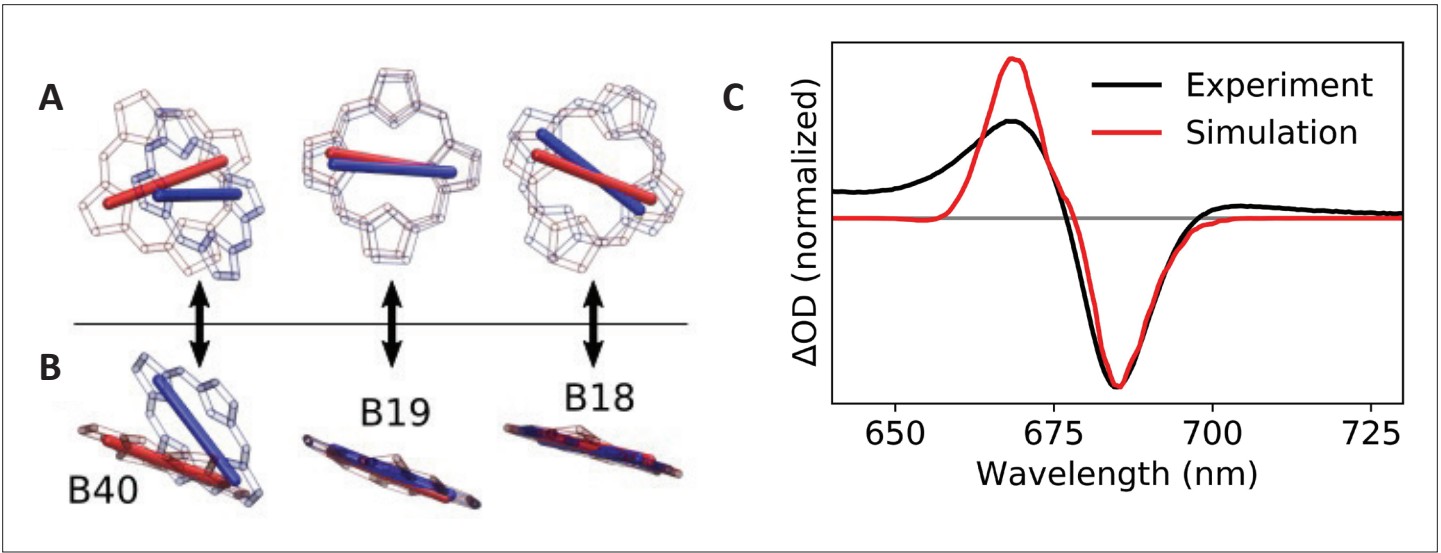

**Figure 5.** Calculated transition dipole vectors for chlorophylls B18, B19, and B40. Viewed from above (**A**) or beside (**B**) the plane of pigment B19. The structures are aligned relative to the main ring atoms of chlorophylls B18 and B19. Red atoms/dipoles refer to *WT Synechocystis*, while blue atoms/dipoles refer to the *Red_c* mutant. (**C**) Simulated (red curve) *Red_c – WT Synechocystis* absorption difference spectra compared with the corresponding experimental spectrum (black curve).

pigment B40 is rotated (predominantly out of plane) by approximately 40° in the *C. aponinum* structure relative to *WT Synechocystis*.

To evaluate the impact of these changes on the $Q_y$ absorption spectrum, we performed excitonic structure calculations using the TrESP coupling and dipole parameters for each structure using the PigmentHunter app at nanoHUB.org (*Safa et al., 2021*). Since no quantitative method exists for assigning pigment site energies based on the structural data, we chose the average site energy values for each pigment to achieve reasonable agreement with the experimental absorption difference spectrum (*Red_c – WT Synechocystis*). The same site energy values (reported on the diagonal of *Table 2*) were used for both complexes; only the dipole moments and coupling values were altered according to the TrESP values calculated from the *C. aponinum* and *WT Synechocystis* structures. Each $Q_y$ absorption spectrum was calculated as an average over 1,000,000 iterations with site energies for each pigment sampled randomly from a 300 cm$^{-1}$ (full width at half-maximum) Gaussian around the respective average value. The final spectrum was convolved with a 10 cm$^{-1}$ Gaussian (full width at half-maximum) for visualization. As seen in *Figure 5C*, the calculated spectrum is in excellent qualitative agreement with the experimental data. (The lack of absorption intensity near and above 650 nm in the calculated spectrum is due to the absence of coupling to vibrational modes in our model.) Since the site energies are chosen in these calculations to match experimental data, these results do not, of course, imply that pigment site energies are identical in the *WT Synechocystis* and *Red_c* complexes. They do, however, demonstrate that the modified coupling values and dipole orientations reflected in the structural data are sufficient to explain the observed spectroscopic changes.

Theoretical modeling of the *Red_d* mutant is more difficult than *Red_c* for two reasons. First, Ca$^{2+}$ removal may induce changes in protein structure (as evidenced by the modified monomer/trimer ratio noted above), which could modify the site energies and couplings of multiple pigments. Second, Ca$^{2+}$ removal almost certainly produces significant site-energy shifts for at least the five pigments closest to the Ca$^{2+}$ binding site (B6, B7, A31, A32, and L3); Chl site energy prediction is, in general, very challenging due to the large number of factors that contribute to environment-induced frequency shifts (e.g., local electrostatics, pigment deformation, electron induction effects, etc.) (*Müh and Zouni, 2020*; *Renger and Müh, 2013*; *Curutchet and Mennucci, 2017*; *Lahav et al., 2021*). For these reasons, we leave detailed modeling of the *Red_d* mutant for a future study.

## Comparing the local environment of chlorophylls between *C. aponinum* and *WT Synechocystis*

Previous theoretical calculations have suggested that the protein electrostatic environments are significant factors in shifting the absorption wavelengths of chlorophylls in addition to strong coupling to neighboring chlorophylls (*Saito et al., 2020*; *Adolphs and Renger, 2006*; *Fiedor et al., 2008*). This led us to examine the structural features that could result in different local electrostatic environments of key chlorophylls between *C. aponinum* and *WT Synechocystis*. To find these features, all chlorophyll rings within 4 Å of an amino acid difference between *C. aponinum* and *WT Synechocystis* were selected. This selection was then compared to predicted LWC sites resulting in the identification of chlorophylls A32 and B7 (*Figure 6A*; *Jordan et al., 2001*). Interestingly, the *WT Synechocystis* PSI structure reveals a vastly different environment around chlorophyll B7 compared to *C. aponinum*. In *WT Synechocystis*, two indole groups of tryptophan residues (I-W20 and L-W65) are located 3.1 Å and 3.9 Å from chlorophyll B7, both along the $Q_y$ dipole axis (*Figure 6B*). However, the structure of *C. aponinum* reveals a leucine and a phenylalanine (PsaI-L27 and PsaL-F64), respectively, at these positions (*Figure 6B*). Because the removal of the Ca$^{2+}$ ion does not fully explain the spectral differences between *C. aponinum* and *WT Synechocystis* above 700 nm, the absence of the indole electrons could be responsible for the remaining difference in LWC content between *C. aponinum* and *WT Synechocystis*.

Additionally, due to its proximity to the B7-A32 dimer, we think chlorophyll A31 could also affect the LWC in *WT Synechocystis*. PsaA-F446 in *WT Synechocystis* is changed to a tryptophan in *C. aponinum* (PsaA-W455) providing an indole group 3.2 Å from A31 along the extended $Q_y$ dipole in *C. aponinum*. Similarly to B7, the presence of the indole group would allow for electron donation to the chlorophyll, altering the electrostatic environment and potentially shifting this chlorophyll absorbance.

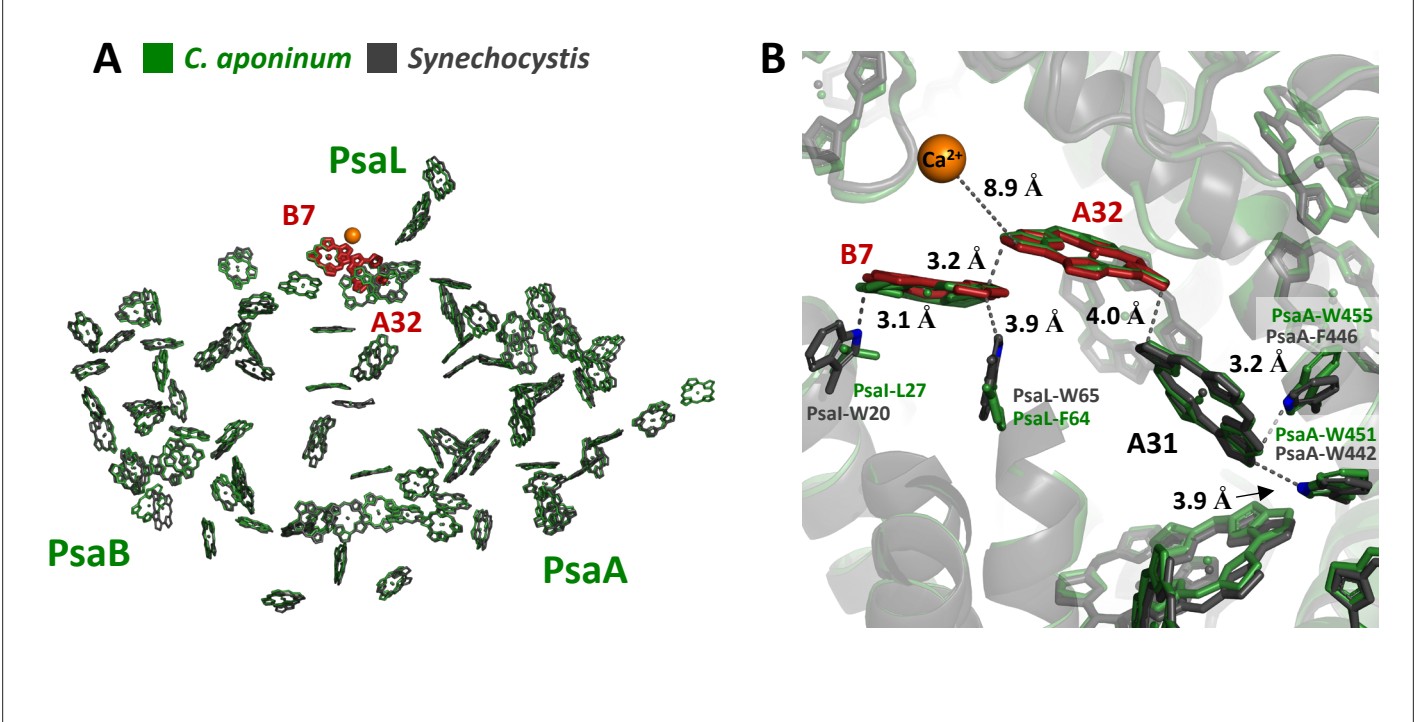

**Figure 6.** Local protein environment of predicted LWC. (**A**) The location of chlorophyll B7 and A32 within each monomer, *C. aponinum* (green), and *WT Synechocystis* (gray, LWC in red). (**B**) The surrounding environment for chlorophyll B7 and A32 with *C. aponinum* (green) and *WT Synechocystis* (gray, LWC in red).

The online version of this article includes the following source data and figure supplement(s) for figure 6:

**Figure supplement 1.** Overlay of *C*.

**Figure supplement 2.** PsaI20 amino acid frequency among 320 bacteria sequences (left) and 581 eukaryote sequences (right).

**Figure supplement 2—source data 1.** Source data for *Figure 6—figure supplement 2*.

**Figure supplement 3.** PsaL65 amino acid frequency among 680 bacteria sequences (left) and 459 eukaryote sequences (right).

**Figure supplement 3—source data 1.** Source data for *Figure 6—figure supplement 3*.

**Figure supplement 4.** PsaA446 amino acid frequency among 774 bacteria sequences (left) and 356 eukaryote sequences (right).

**Figure supplement 4—source data 1.** Source data for *Figure 6—figure supplement 4*.

## Discussion

The unique environment that *C. aponinum* was isolated from is exposed to high light and a constant supply of moisture. In deserts where water is scarce, cyanobacteria are mostly active after rainstorms, when there is cloud cover, or around dawn when dew is present, both low-light environments (*Harel et al., 2004*; *Potts, 1994*). Under drought conditions, cyanobacteria have adapted to decrease PSI and PSII activity, protecting cells from desiccation (*Satoh, 2002*; *Lawlor, 2002*; *Potts and Friedmann, 1981*; *Chaves, 1991*). The photoinactivation caused by drought may have prevented any evolutionary pressure toward tolerating high light (*Harel et al., 2004*; *Potts, 1994*). This makes *C. aponinum* a unique candidate to study adaptations in cyanobacteria to survive in high-light conditions.

*C. aponinum* was isolated from environmental samples of a microbial mat and grown under ~3200 µmol photons m$^{-2}$s (*Bekker, 2004*) in the lab, greatly exceeding the maximum amount of solar irradiance it would naturally experience. Strains of *C. aponinum* were already known to grow in high temperatures, and now it is shown that they grow in high light, demonstrating the innate ability of this organism to survive in extreme environments. When exposed to high light, *C. aponinum* increases its carotenoid content as well as changing the distribution of excitation energy between PSI and PSII to favor PSI, suggesting that PSI is important for growth under high light.

Purified PSI from *C. aponinum* is found as a trimer and its subunit composition is identical to *WT Synechocystis*. Protein sequence alignments identified insertions in PsaB and PsaL, as well as a

drastically different C-terminus of PsaL. Absorbance measurements show *C. aponinum* PSI absorbs less in the far-red than *WT Synechocystis* PSI, indicating a lower content of LWC in *C. aponinum*. Gaussian deconvolutions indicate that these LWC contribute to absorbance bands at 704 nm and 711 nm. Surprisingly, the 77 K emission peak is shifted 2 nm red compared to *WT Synechocystis*. This is a seemingly contradictory result, as *C. aponinum* contains less LWC, which are the main emitters of the 77 K emission. We suggest that the elimination of some LWC in *C. aponinum* cause a greater emission from a different either C706 or C714 (*Toporik, 2020*; *Khmelnitskiy et al., 2020*; *Riley et al., 2007*) LWC with a lower energy. This is consistent with the findings that there is less red absorption in *C. aponinum* together with the red shifted emission at 77 K.

To understand the structural features that could explain these spectroscopic differences, the structure of PSI was determined by cryo-EM. The structure was similar to trimeric PSI structures from cyanobacteria with a few differences. One of which being an insertion in PsaB manifests as a loop on the stromal side of the protein near three chlorophyll molecules, chlorophylls B18/B19/B40.

The Insertion in PsaB causes chlorophyll B40 to alter its orientation in comparison to *WT Synechocystis* (*Figure 3B*). To test whether the orientation of this chlorophyll is responsible for differences in the spectroscopic properties, we constructed a chimeric PSI, *Red_c*, with this insertion in *WT Synechocystis*. *Red_c* demonstrates that insertion of the loop results in a blue shift within the $Q_y$ transition attributed to the change in chlorophyll B40 position. Although the difference spectrum shows a minimum absorbance of 685 nm (*Figure 4D*), occupying an energetic position lower than most of the antenna chlorophylls, chlorophyll B40 is not a LWC (in *WT Synechocystis*) according to the accepted definition.

Mutating the $Ca^{2+}$ coordinating residue in PsaL to mimic the *C. aponinum* sequence resulted in a decrease in absorption at 704 nm, indicating that the $Ca^{2+}$ ion tunes LWC in *WT Synechocystis* and its absence in *C. aponinum* probably contributes to the observed loss of LWC absorption. A similar effect of $Ca^{2+}$ binding was shown previously in the light harvesting complex-1 of purple bacteria (*Kimura et al., 2008*; *Swainsbury et al., 2017*). The $Ca^{2+}$ ion in PsaL of *WT Synechocystis* is located near known LWCs, chlorophylls B7 and A32 (*Khmelnitskiy et al., 2020*; *Jordan et al., 2001*), and our results support its contribution to the low-energy states of this chlorophyll pair (*Figure 4D,E*). We do not resolve any changes in the 77 K emission between *WT Synechocystis* and *Red_c* or *Red_d*, this is in agreement with previous results showing that red shifting the B7-A32 dimer contributes very little to the maximum emission from the PSI trimer in *WT Synechocystis,* even at 4 K (*Khmelnitskiy et al., 2020*). This suggests that the terminal emitter for PSI is a different LWC.

The *Red_d* mutation does not account for all the changes in LWC absorption seen between *C. aponinum* and *WT Synechocystis*. We suggest that the indole groups of tryptophan residues proximal to chlorophylls B7–A32–A31 also contribute to the spectral differences observed between PSI from *C. aponinum* and *WT Synechocystis*. The lone pair from the tryptophan indole group could affect the HOMO/LUMO energy levels of these chlorophylls, thus changing the site energies (*Saito et al., 2020*). In the case of chlorophylls B7–A32–A31 in *C. aponinum*, the combination of losing PsaL-W20 and PsaL-W65 to uncharged species and gaining an indole lone pair in PsaA-W455 compared to *WT Synechocystis* would change the local environment along the transition dipole of this chlorophyll aggregate (*Figure 6*; *Saito et al., 2020*). Interestingly, similar amino acid variations are also seen in PSI structures isolated from plants, green algae, red algae and diatoms, presumably altering the local environment in a similar way throughout the evolution of photosynthesis in eukaryotic organisms (*Figure 6—figure supplement 1* and *Figure 6—figure supplements 3 and 4*). These differences around LWC chlorophylls probably account for additional differences in the absorption spectra between *C. aponinum* and *WT Synechocystis*. To confirm this and to quantify the effect of each tryptophan residue on these chlorophylls, additional studies are needed.

The orientation of the PsaL C-terminus in *C. aponinum* differs from other structures of trimeric PSI from cyanobacteria (but is similar to *Synechococcus; Cao et al., 2020*). As previously reported, the deletion of PsaL or the addition of a terminal histidine to the C-terminus PsaL in *WT Synechocystis* leads to the complete dissociation of trimeric PSI, underlining the importance and sensitivity of PsaL to trimerization (*Malavath et al., 2018*; *Netzer-El et al., 2018*). In *C. aponinum*, the C-terminus is shifted away from the adjacent monomer and there is no $Ca^{2+}$ ion observed in the density of the structure (*Figure 4*). High resolution structures of PSI showed that the C-terminus of PsaL interacts with an adjacent monomer through a $Ca^{2+}$ ion coordinated by an aspartic acid (*Netzer-El et al., 2018*; *Mazor*

*et al., 2015*). We now show that Ca²⁺ contributes but is not essential for trimer assembly. Unexpectedly, we also show that Ca²⁺ binding also tunes and red-shifts LWC in PSI. This agrees with the fact that in monomeric PSI from eukaryotes, a Ca²⁺ ion is not seen (*Mazor et al., 2015*), together with the lower LWC content of eukaryotic core complexes (*Croce et al., 1998*). In addition, upon monomerization, the cyanobacterial PSI loses LWC absorption, as monomerization is also accompanied by the loss of Ca2+, our findings suggest that Ca2+ loss contributes to these differences (*Gobets et al., 2001*).

How prevalent is Ca²⁺ binding in PSI? Comparing prokaryotic and eukaryotic PsaL shows differences in conservation in Ca²⁺ coordinating residue in cyanobacteria. This residue exhibits high conservation as an asparagine across eukaryotes (*Figure 3—figure supplement 6*), while an alignment of 680 cyanobacterial PsaL sequences revealed that despite the high conservation around the Ca²⁺ coordinating residue in prokaryotes, this specific residue is variable in cyanobacteria (*Figure 3—figure supplement 5*). Residues containing a negative charge are seen at this location in ~50 % of species, including *WT Synechocystis* and *T. elongatus*, which likely corresponds to coordination of Ca²⁺. In *Synechococcus* however, the coordinating residue is an asparagine. The recently solved PSI structure from *Synechococcus* lacks a Ca²⁺ ion, giving further support to the importance of this residue in coordinating a calcium ion (*Cao et al., 2020*). While the presence of Ca²⁺ aids in the stabilization of trimerization for some species, the structure of trimeric PSI from *C. aponinum* and *Synechococcus* shows it is not required. It is interesting to note that the majority of differences in sequence between *C. aponinum* and *WT Synechocystis* occur in the intermonomer space. These differences probably stabilize the monomer–monomer interaction in *C. aponinum* despite the absence of the PsaL Ca²⁺ ion.

The mechanism underlying the contribution of PSI to photoprotection in *C. aponinum* is not presently clear; however, PSI is potentially the most potent quencher of excitation energy in the cell. The optical properties of PSI from *C. aponinum* are similar to the eukaryotic core PSI family, with a low number of LWC. The contribution of PSI to cellular resistance to high light can stem from its interaction with other components of the light reactions, such as PSII, PBS, or the stress-induced antenna, IsiA, as has been suggested in the past (*Tiwari, 2016*). The role played by LWC in these interactions is not clear to a large degree because specific mutations or precise information on the architecture of LWC in PSI is scarce. Previously, it was hypothesized that LWCs can serve several functions within the core PSI antenna. Light harvesting in the far-red region is one clear suggestion, but it was also shown that low energy states associated with LWC are quenched by an oxidized P700⁺ and that this varies between cyanobacterial species (*Schlodder et al., 2005*; *Herascu et al., 2016*; *Schlodder et al., 2011*). This shows that energy transfer in the core PSI antenna varies depending on the oxidation state of P700 and this can be important during high-light conditions when P700 is in its oxidized state. A specific LWC in PSI, in close proximity to the ETC (*Schlodder et al., 2005*; *Herascu et al., 2016*; *Schlodder et al., 2011*), is responsible for this effect and can play a role in photoprotection. In this scenario, the optical adaptations that are seen in *C. aponinum* facilitate energy transfer through this LWC, this is consistence with our 77 K emission results, which show red shifted emission from *C. aponinum* compared to *WT Synechocystis*, probably due to the contribution from this site. Overall, it is necessary to develop an understanding of individual LWC as specific sites (these sites probably include more than one chlorophyll molecule) that carry out specific functions. An essential step on this route is identifying specific LWC and factors that tune them in PSI, as the present study does.

The environment that *C. aponinum* was isolated from provides ideal conditions for using P700 as a photoprotective mechanism. PSI-specific photodamage is known to be induced by fluctuating light and cold temperatures (*Sonoike, 2010*; *Terashima et al., 1994*; *Sonoike et al., 1997*); however, the environment *C. aponinum* was isolated from rarely experiences these conditions. Constant light and temperature may allow P700 to withstand more irradiance without causing photoinhibition.

PSI displays high conservation in sequence and structure across domains despite the vastly different environments occupied by photosynthetic organisms. The results of this work show that small structural variations can have large effects, highlighting the sensitivity of pigments to the local electronic environment, and potentially giving rise to physiological advantages (*Wientjes et al., 2012*; *Wientjes et al., 2011*). The structural effects on spectroscopic properties of PSI observed in this work lay the foundations for intelligently designing photosynthetic organisms with absorption spectra tuned to specific light environments. The structure and biochemical characterization of trimeric PSI from *C. aponinum* advances the fundamental understanding of the photosynthetic machinery in organisms

that can survive extreme light conditions and reinforces the need to study extremophiles and their adaptations to fully understand the photosynthetic process.

## Materials and methods

### Selection conditions

Crude biofilm samples were placed in 50 ml of BG-11 media in Erlenmeyer flasks and exposed to >3000 µmol photons $m^{-2}s$ (*Bekker, 2004*). Samples were not stirred. Once noticeable growth had occurred, about 2 weeks, samples were agitated by aggressively swirling growth flasks. One milliliter was then plated onto a BG-11 agar plate and allowed to grow. Single colonies were picked and continually streaked to achieve a pure culture of the photosynthetic organism.

### 16S rRNA sequencing

A sterile culture of *C. aponinum* was grown and harvested mid log phase growth. Genomic DNA was extracted by a modified phenol chloroform extraction as previously described (*Adolphs and Renger, 2006*). Using primers designed by *Li et al., 2014*, the 16 S rRNA gene was amplified and sent for sanger sequencing.

### Genomic DNA sequencing

Illumina compatible Genomic DNA libraries were generated on the Apollo 384 liquid handler using KAPA Biosystem's LTP library preparation kit (KK8232). DNA was sheared to approximately 600 bp fragments using a Covaris M220 ultrasonicator, end repaired, and A-tailed as described in the Kapa protocol. Illumina-compatible adapters with unique indexes (IDT #00989130v2) were ligated on each sample individually. The adapter ligated molecules were cleaned using Kapa pure beads (Kapa Biosciences, KK8002) and amplified with Kapa's HIFI enzyme (KK2502). Each library was then analyzed for fragment size on an Agilent's Tapestation and quantified by qPCR (KAPA Library Quantification Kit, KK4835) on Thermo Fisher Scientific's Quantstudio 5. Libraries were then multiplexed and sequenced on 2 × 250 flow cell on the MiSeq platform (Illumina) at the ASU's Genomics Core facility.

### Genomic analysis

The raw Illumina MiSeq 2 × 250 bp reads (14,485,288 pairs of reads) were quality checked using FastQC v0.10.1, followed by adapter trimming and quality clipping by Trimmomatic 0.35. Any reads shorter than 150 bp were dropped. Any reads with start, end, or the average quality within 4 bp window falling below quality scores 18 were trimmed. A clean 12,273,912 read pairs survived for further insert size estimation. Kmer analysis was ran by Jellyfish 2.2.4 over both entire 14,485,288 read pairs and clean 12,273,912 read pairs for genome size estimation. Cleans reads were aligned to *C. aponinum* PCC 10605 (cyanobacteria) reference genome (https://www.ncbi.nlm.nih.gov/assembly/GCF_000317675.1/) by bwa mem 0.7.15 for insert size estimation. Spades 3.7.1 with mismatch corrector mode was applied for whole-genome assembly with kmer size 21,33,55,77,99,127. Best whole genome assembly with kmer size 127 was evaluated by comparing to *C. aponinum* PCC 10605 reference genome by Quast 4.5 (http://bioinf.spbau.ru/quast). When sorting contigs from largest to smallest, first 80 contigs with minimum length 1000 bp were extracted. CAR, a novel reference-based contig assembly and scaffolding tool (http://genome.cs.nthu.edu.tw/CAR/), was applied on the 80 contigs for scaffolds. In order to improve assembly, SSPACE was applied to scaffold pre-assembled contigs using NGS paired-read data. Eighty contigs were kept in final 55 scaffolds. Quast was used to evaluate assembly with 55 scaffolds. BUSCO, a tool for assessing genome assembly and annotation completeness with benchmarking universal single copy orthologs (http://busco.ezlab.org/), indicated 97.3 % genome completeness with 812 complete BUSCOs out of 834 total BUSCOs defined in cyanobacteria database. Total of 989 genes were predicted de novo by Glimmer (Gene Locator and Interpolated Markov Model ER, https://ccb.jhu.edu/software/glimmer/). Additional genome annotation was performed by protein homology-based tblastn (blast +2.3.0) approach using protein sequences of *C. aponinum* PCC 10605 reference genome. Three thousaand three hundred and thirty-three hits were identified. Genomic data was deposited to NCBI (NCBI:txid2676140).

### Culture conditions

*C. aponinum* used for the structural studies was cultured in BG11 medium supplemented with 6 µg/ml ferric ammonium citrate under continuous white light (~40 µmol photons $m^{-2}s^{-1}$) in 30 °C.

## Growth tests

*C. aponinum* and *Synechocystis* sp. PCC 6803 cells were cultured in BG11 liquid medium supplemented with 6 µg/ml ferric ammonium citrate under continuous white light (~40 µE) in 30 °C. The optical density was adjusted to 5 at 730 nm for *Synechocystis* and three for *C. aponinum*. Each culture was diluted ×5, ×25, ×125, ×625, ×3125, ×15,625 and grew on BG11 plates in different light intensities to determine cell viability.

## Thylakoid preparation

Cells were harvested during log phase growth by centrifugation at 12,000 rpm for 3 min at room temperature. Cells were washed in STN1 buffer (30 mM Tricine–NaOH pH 8, 15 mM NaCl, 0.4 M sucrose) and pelleted again to remove any excess growth media, then lysed in a cooled Constant Cells Disruptive Systems French press for three cycles at 30,000 psi. The lysate was cleared of cell debris by centrifuging 12,000 rpm for 5 min in a F20–12 × 50 LEX rotor (Thermo Scientific). Membranes present in supernatant were then pelleted by ultracentrifugation (Ti70 rotor) for 2 hr at 45,000 rpm and 4 °C. Membranes were then resuspended in STN1 with 150 mM NaCl and allowed to incubate on ice for 15 min before ultracentrifugation (Ti70 rotor) for 2 hr at 45,000 rpm and 4 °C. The pellet was then resuspended in 15 ml of STN1 and stored at –80 °C.

PSI purification n-dodecyl β-maltoside (DDM, Glycon) was added to the membrane stock to achieve a ratio of 15:1 DDM-to-chlorophyll ratio, and the samples were manually mixed a few times then allowed to incubate for 30 min on ice. Membranes were centrifuged for 30 min at 45,000 rpm at 4 °C to remove any insoluble material. The supernatant was then loaded onto a diethylaminoethyl column (toyopearl DEAE 650 C) and eluted with a linear NaCl gradient (15–350 mM) in 30 mM Tricine–NaOH pH 8, 0.2 % DDM. The dark green peak corresponding to the PSI trimer was collected and precipitated with 8 % PEG3350 (Hampton Research) and 150 mM NaCl. This was then centrifuged at 5000 rpm for 5 min at 4 °C and the supernatant discarded. The pellet was resuspended in 30 mM Tricine–NaOH pH 8, 15 mM NaCl, 0.1 % DDM and loaded onto a 10–30% sucrose gradient (30 mM Tricine–NaOH pH 8, 75 mM NaCl, 0.05 % DDM). This was centrifuged (Beckman SW40 rotor) for 16 hr at 36,000 rpm. The dark green band was collected and used for subsequent experiments.

## Absorption and fluorescence spectroscopy

Absorption spectra were recorded on a Cary 4,000 UV–Vis spectrophotometer (Agilent Technologies). Fluorescence spectra were recorded on a Fluoromax-4 spectrofluorometer (HORIBA Jobin-Yvon). The slit width was set to 5 nm on both the entrance and exit monochromators for room temperature measurements. For 77 K measurements, slit width of 5 nm and 3 nm were used for the entrance and exit monochromators, respectively. Samples were diluted to an optical density of 1 and 0.1 at 680 nm for absorption and fluorescence measurements respectively, using buffer containing 30 mM Tricine–NaOH pH 8, 15 mM NaCl, and 0.05 % β-DDM. The resulting spectra were normalized to the area of the chlorophyll Q bands between 550 and 775 nm. Whole-cell measurements were performed using an integrating diffuse reflectance sphere (DRA 900) to correct for scattering by the cells. For 77 K fluorescence measurements, samples were adjusted to an OD680 of 0.1 in a buffer of 50 % glycerol 30 mM tricine pH 8.0, 15 mM NaCl, and 0.02 % β-DDM. An Oxford instruments Cryostat was used to cool the sample to 77 K (cells were plunged into liquid nitrogen and measured immersed in liquid nitrogen). Figures were prepared using OriginPro (OriginLab).

## Sample preparation for single particle cryo-EM

The PSI band from the sucrose gradient was collected, NaCl was added to a final concentration of 150 mM and the complex was precipitated using 9 % PEG3350. After centrifugation (5000 rpm, 5 min in an Eppendorf tabletop), the green precipitate was resuspended in buffer (30 mM Tricine–NaOH pH 8, 150 mM NaCl, and 0.02 % DDM), and any undissolved material was removed by repeating the centrifugation step (14,000 rpm, 5 min). The chlorophyll concentration in the soluble material was adjusted to 1.2 mg/ml using the above buffer. Three microliters of the PSI complex was added to holey carbon grids (C-flat 1.2/1.3 Cu 400 mesh grids [Protochips, Raleigh, NC]) after soaking the grids in buffer. The sample was vitrified by flash plunging the grid into liquid ethane using manual plunger with blotting time of 6 s. The grids were stored in liquid nitrogen until data collection.

## Data acquisition

The cryo-EM specimens were imaged on a Titan Krios transmission electron microscope (Thermo Fisher - FEI, Hillsboro, OR). The electron images were recorded using a K2 Summit direct electron detect camera (Gatan, Pleasanton, CA) in super-resolution counting mode. Image collection was automated with SerialEM (*Fiedor et al., 2008*) utilizing scripting of stage shifts between hole exposures. The defocus was set to vary between 0.8 and 2.6 µm, corresponding to a super-resolution pixel size of 0.525 Å at the specimen level. The counting rate was adjusted to 7.614 $e^-$/Å$^2$ s. Total exposure time was 8 s accumulating to a dose of 61 $e^-$/Å$^2$.

## Data processing

A flow chart describing data handling is shown in *Figure 3—figure supplement 1*. MotionCor2 (*Harel et al., 2004*) was used to register the translation of each sub-frame, and the generated averages were Fourier-cropped to 1.5 times and dose-weighted (*Potts, 1994*). CTF parameters for each movie were determined using CTFFIND4 (*Satoh, 2002*). Relion was then used for the subsequent data processing (*Lawlor, 2002*). A set of manually picked particles (~1000) from an early data set was subjected to a few rounds of unsupervised 2D classification and then used to generate an initial 3D volume. This volume was then used on a later data set as the template for the automated particle picking procedure as implemented in Relion which yielded 256,410 particles. This particle set was subjected to several rounds of unsupervised 2D classification (Relion) resulting in a set of 132,677 particles. This particle set was then subjected to a focused 3D classification on the PsaL subunit. This procedure yielded eight classes with one dominate class. This class was selected yielding 73,984 particles. 3D refinement (C3 symmetry) using this set yielded a volume at a resolution of 3.71 Å. CTF refinement was used (*Potts and Friedmann, 1981*), followed by 3D refinement (C3 symmetry), yielding the final resolution of 3.79 Å. Particle polishing was implemented followed by 3D refinement (C3 symmetry) yielding a resolution of 3.75 Å. This was followed by three cycles of CTF refinement and 3D refinement (C3 symmetry), yielding a resolution of 2.88 Å. The detergent signal was subtracted from the PSI trimer, followed by a C3 expansion of the particles, yielding a total particle count of 221,952. This particle set was used for a 3D refinement (C1 symmetry) resulting in a map of 3.0 Å. Multibody refinement was used to generate the final map, of 2.74 Å according to the gold standard FSC criteria (*Chitnis and Chitnis, 1993*). The final map was sharpened using the postprocessing procedure in Relion, and the b-factor used for sharpening was –72.48. Local resolution was estimated using ResMap.

## Model building and refinement

The initial PSI model was taken from the 2.5 Å x-ray structure of the trimeric PSI from *Synechocystis* (PDBID: 5OY0) (*Toporik, 2020*). The model was docked into the map using PHENIX (*Chaves, 1991*). The final model was refined against the cryo-EM density map using phenix.real_space_refine (*Riley et al., 2007*; *Kimura et al., 2008*). Final model statistics are shown in *Table 1*, and side chain resolvability was calculated using MapQ (*Swainsbury et al., 2017*; *Supplementary file 1a*). PyMOL (*Jordan et al., 2001*) and UCSF Chimera (*Mazor et al., 2015*) were used to generate all images.

## Strain construction and growth

### *Red_c* construction

A plasmid (p60) containing the entire PsaAB operon marked with Kanamycin resistance gene was previously constructed (*Tikkanen et al., 2014*). The *Red_c* mutation was constructed into p60 by adding the loop sequence observed in *C. aponinum* using the p60_red_c_Forward_insert and P60_red_c_Reverse_insert primers and p60 as a template. The two fragments were assembled using the NEBuilder HiFi DNA Assembly Master Mix. All plasmids were sequenced before being used to transform *Synechocystis* sp. PCC6803 according to standard protocols. Complete and correct replacement of PsaB was verified by PCR and sequencing.

### *Red_d* construction

A plasmid (PsaLPsaI) containing both the PsaL and PsaI genes marked with a chloramphenicol resistance gene 0.1 kb upstream to PsaI was constructed from four PCR amplified fragments (*Tikkanen et al., 2014*). PsaL and PsaI fragments, with up an 0.8 kb upstream fragment, were amplified using the PsaL_R and PsaL_F primer pairs from the *Synechocystis* sp. PCC6803 genome. The chloramphenicol

resistance gene was amplified from previously constructed plasmids in our lab, using Cm_F and Cm_R primer pairs. A 0.8 kb downstream fragment from PsaI was amplified using Down_F and Down_R primer pairs. A pJET backbone was amplified using the primer pairs Backbone_F and Backbone_R. The four fragments were assembled using the NEBuilder HiFi DNA Assembly Master Mix. The *Red_d* mutation was constructed into PsaLPsaI by creating a point mutation observed in *C. aponinum* using the Ca_D2L_F/PsaL_R and Ca_D2L_R/PsaL_F primer pairs and PsaLPsaI as a template. The two fragments were assembled using the NEBuilder HiFi DNA Assembly Master Mix. All plasmids were sequenced before being used to transform *Synechocystis* sp. PCC6803 according to standard protocols. Complete and correct replacement of all aspects was verified by PCR and sequencing. All primer sequences are listed in *Supplementary file 1b*.

## Excitonic structure calculations

Electronic transition dipoles and coupling elements amongst Chls B18, B19, and B40 for *WT Synechocystis* were calculated using the TrESP method (*Moro et al., 2007*). The PDB: *6UZV* model of trimeric PSI was used as *WT Synechocystis,* and the *C. aponinum* PSI structure (PDB:6VPV) was used to approximate the Red_c structure. For the TrESP calculation, we used the gas-phase transition charges previously calculated (*Moro et al., 2007*) and rescaled here to ensure that each pigment carried a $Q_y$ dipole moment strength of 4.3 Debye (*Malavath et al., 2018*; *Mazor et al., 2013*). Excitonic structure calculations using the TrESP coupling and dipole parameters for each structure using the PigmentHunter app at nanoHUB.org (*Winckelmann et al., 2015*). Site energy values for each pigment were varied to achieve reasonable agreement with the experimental absorption difference spectrum (*Red_c - WT Synechocystis*). The same site energy values (reported on the diagonal of *Table 1*) were used for both complexes; only the dipole moments and coupling values were altered according to the TrESP values calculated from the *C. aponinum* and *WT Synechocystis* structures. Each $Q_y$ absorption spectrum was calculated as described previously (*Mazor et al., 2013*) as an average over 1,000,000 iterations with site energies for each pigment sampled randomly from a 300 cm⁻¹ (full width at half-maximum) Gaussian around the respective average value. The final spectrum was convolved with a 10 cm⁻¹ Gaussian (full width at half-maximum) for visualization; no vibrational or phonon-sideband structure was included.

## Additional information

### Funding

| Funder | Grant reference number | Author |
|---|---|---|
| National Institute of Food and Agriculture | 2020-67034-31742 | Zachary Dobson |
| Biodesign, Center of Applied Structural Discovery | | Zachary Dobson |

The funders had no role in study design, data collection and interpretation, or the decision to submit the work for publication.

### Author contributions

Zachary Dobson, Conceptualization, Data curation, Formal analysis, Funding acquisition, Investigation, Methodology, Project administration, Resources, Software, Validation, Visualization, Writing - original draft, Writing - review and editing; Safa Ahad, Data curation, Formal analysis, Investigation, Methodology, Software, Writing - original draft, Writing - review and editing; Jackson Vanlandingham, Investigation; Hila Toporik, Formal analysis, Investigation, Methodology, Software, Supervision, Visualization, Writing - review and editing; Natalie Vaughn, Conceptualization; Michael Vaughn, Conceptualization, Methodology; Dewight Williams, Investigation, Methodology; Michael Reppert, Conceptualization, Methodology, Resources, Software, Supervision, Visualization, Writing - original draft, Writing - review and editing; Petra Fromme, Conceptualization, Funding acquisition, Methodology, Resources, Supervision, Writing - original draft, Writing - review and editing; Yuval Mazor, Conceptualization, Data

curation, Formal analysis, Funding acquisition, Investigation, Methodology, Project administration, Resources, Software, Supervision, Validation, Visualization, Writing - original draft, Writing - review and editing

### Author ORCIDs
Zachary Dobson ⓘD http://orcid.org/0000-0003-4951-1701
Michael Vaughn ⓘD http://orcid.org/0000-0001-9357-094X
Yuval Mazor ⓘD http://orcid.org/0000-0001-5072-0928

### Decision letter and Author response
Decision letter https://doi.org/10.7554/eLife.67518.sa1
Author response https://doi.org/10.7554/eLife.67518.sa2

## Additional files

### Supplementary files
• Supplementary file 1. (1a) Individual chains and ligand resolvability according to Q-scores. Scores were calculated using the MapQ plugin in UCSFChimera. (1b). Primer list used for mutant construction.

• Transparent reporting form

### Data availability
The final model (PDBID 6VPV) and map (EMD-21320) were deposited in the Protein Databank and Electron Microscopy Database, respectively. C. aponinum genomic DNA was deposited in NCBI genebank under NCBI:txid2676140.

The following dataset was generated:

| Author(s) | Year | Dataset title | Dataset URL | Database and Identifier |
|---|---|---|---|---|
| Dobson Z, Vaughn N, Vaughn M, Fromme P, Mazor Y | 2019 | Cyanobacterium aponinum 0216, whole genome shotgun sequencing project | https://www.ncbi.nlm.nih.gov/bioproject/PRJNA580528 | NCBI BioProject, PRJNA580528 |
| Dobson Z, Vaughn N, Vaughn M, Fromme P, Mazor Y | 2021 | Trimeric Photosystem I from the High-Light Tolerant Cyanobacteria Cyanobacterium Aponinum | https://www.rcsb.org/structure/6VPV | RCSB Protein Data Bank, 6VPV |
| Dobson Z, Vaughn N, Vaughn M, Fromme P, Mazor Y | 2021 | Trimeric Photosystem I from the High-Light Tolerant Cyanobacteria Cyanobacterium Aponinum | https://www.ebi.ac.uk/emdb/entry/EMD-21320 | EMD, 21320 |

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
