## [Decision Letter]

**Acceptance summary:**

The work presents high resolution structure and functional measurements of a photosystem I complex from a high light tolerant species of cyanobacteria. The results give important clues about the adaptations of photosynthetic systems that allow them to function under such harsh environmental conditions. The revised version of the paper clears up most of the questions presented by the reviewers and adds data pointing to a possible new insights on the function of long wavelength chlorophylls in photosystem I light harvesting complexes.

**Decision letter after peer review:**

Thank you for submitting your article "The Structure of Photosystem I from a High-Light Tolerant Cyanobacteria" for consideration by *eLife*. Your article has been reviewed by 4 peer reviewers, one of whom is a member of our Board of Reviewing Editors, and the evaluation has been overseen by Olga Boudker as the Senior Editor. The following individual involved in review of your submission has agreed to reveal their identity: Geoffry A. Davis (Reviewer #2).

All of the reviewers felt that the structural work was both interesting and well done. Also, there was considerable enthusiasm for the approach of comparing structures from diverse species that might reveal interesting adaptations to different environments.

On the other hand, all of the reviewers had substantial questions about how the identified structural variations relate to the biophysical/spectroscopic measurements or whether these differences were relevant to any physiological changes.

It should be possible to address the spectroscopy issues by reanalyzing the data along the lines of that proposed by reviewer 4.

It will probably be more difficult to deal with the issue of physiological relevance. As stated by several of the reviewers, given the many changes that are likely to have occurred between the two species, it is difficult to know if the specific structural variations identified in the paper are important, or indeed, if they have any physiological consequences by themselves. It may be, as pointed out by one reviewer, that the big story is not that there are vast differences, but that there are so few differences.

It was also pointed out that there is an inherent difficulty with comparing light acclimation between two species, especially those that are quite distantly related and grow under very different light regimes.

From these comments, it would seem that directly addressing the question of physiological relevance will require some additional data, though it may be possible to approach to achieve this by extending the bioinformatics approaches rather than additional experimentation.

A second approach would be to better focus the discussion to aspects of the structure that are more directly supported by the data itself, and discuss the potential implications more hypothetically. It would be great if this could be achieved while retaining the intended high impact.

*Reviewer #1:*

The paper can be improved by: (1) better describing the fitting methods and their bases; (2) describing alternative models; (3) providing more evidence and arguments that the structural differences that we observed are relevant to the selection pressure or physiological differences required for growth in the different light regimes; and (4) acknowledging and discussing that the light environments are relative, e.g. that using Synechococcus, which is generally found growing at very low light, may not be an adequate (sole) comparison for a high light species.

1) "…had a constant drip of fresh water and exposed to extreme light conditions." How is 1300 uE an extreme light? Are all terrestrial vascular plants now extremophiles?

2) In this regard, choosing *Synechocystis* as a comparison is a bit of a "straw man" and it would be more appropriate to compare to a wider range of species/strains. In other words, wouldn't it be more accurate to say that *Synechocystis* is adapted to LOW light, rather than *C. aponiunum* adapted to high? If this is true, then the story may be about selection pressure for low light rather than photoprotectiom.

3) The work shows that *C. aponium* can grow at higher light than Synechocystis. However, it was not possible to attribute this difference to PSI properties. This would be a pretty high bar to set, of course, but can the authors make the case that the differences they do observe are relevant?

4) The experiments exposed plates to 3000 µmol photons m-2 s^-1^, which is higher than that these cells experience in the real world. What is the argument that 3000 uE is physiologically relevant? Is it not the case that this selects for something not relevant to nature?

5) The 77K emission spectrum is not a measure of PSI:PSII content, but on the distribution of excitation energy. Further, the interpretation of these spectra is dependent on the specific excitation wavelength. For instance, if cyan light will excite the PBS, which can dump energy differently into PSI and PSII, compared to, e.g. blue light. The emission spectrum is also impacted by cell density and other factors. None of these issues is acknowledged in the text, nor is that excitation wavelength described in the Methods, as is clear from the scant description given in the text: "An Oxford instruments Cryostat was used to cool 26 the sample to 77 K (cells were plunged into liquid nitrogen and measured immersed in liquid nitrogen)." which does not give any indication of the instrument, excitation, cell concentration, etc., though Figure 2 mentions that particular data was collected with excitation at 440 nm. What is the resolution of the spectrometer?

6) Figure 3. The spectra are interpreted as apparent shifts based on subtracting pairs of spectra normalized to their peaks. This seems to be based on a subjective interpretation and other interpretations are possible. Changing the normalization point could result in an entirely different interpretation. For instance, normalizing to a point on the red side of the peak could give the appearance of an increase in absorbance or emission on the blue edge of the spectrum (rather than a shift). The deconvolution of the absorbance spectra using three Gaussian components as in Figure 2D is likewise based on an assumption or interpretation that there are only these components.

7) The text needs to be more circumspect and/or directly address this as a possibility, since such alternative normalization could indicate a quite different story. For example, the text reads: "The difference spectrum of *C. aponinum* – *Synechocystis* shows a strong negative peak at 701 nm revealing that *C. aponinum* PSI contains less LWC 150 than *Synechocystis* (Figure 2D)." It suggests this, but there are other interpretations. This issue is more evident further in the manuscript, in Figure 4, a different method of normalization was used, as stated in the following: "The absorbance spectra were normalized to the area between 650 nm to 775 nm and the spectrum of Wild Type *Synechocystis* was 238 subtracted from Red_c (Figure 4C and 4D)." As argued above, changing the normalization method can give rise to different interpretations, so some real justification must be presented for using a particular approach, and an even stronger argument needs to be made when changing the normalization between samples. None is given.

8) Figure 4 D compares isolated complexes from two different species and the differences spectra can be interpreted as a series of narrowing and shifting of spectra, but all of which are likely to be dependent on the method of normalization, which is not adequately described. Moever, the method used to make the differences spectra were not consistently applied, e.g. "[t]he absorbance spectra were normalized to the area between 650 nm to 775 nm and the spectrum of Wild Type *Synechocystis* was subtracted from Red_c (Figure 4C and 4D)." What were the criteria for using the different procedures and how was bias avoided?

9) The shifts in absorbance and fluorescence emission are quite small, especially realizing that the bands tend to sharpen at 77K. At RT, there may be essentially no functionally-relevant differences in the spectra. How can this be addressed?

10) Can the text be expanded to further discuss the very large apparent difference in PsaF?

11) *Cyanobacterium aponinum* (C. 78 aponinum). should be both Ital. Is this an official name?

*Reviewer #2:*

The manuscript by Dobson et al., describes photosystem I (PSI) of a high-light tolerant cyanobacterium, *Cyanobacterium aponinum* 0216, isolated from an environmental sample at both the sequence and structural level and attempts to test how differences in the *C. aponinum* (PSI) sequences may result in the structural and spectroscopic differences characterized. The authors solved the Cryo-EM structure of trimeric PSI of *C. aponinum* and noted two major differences with the model cyanobacterium *Synechocystis* PCC 6803 trimeric PSI, namely insertions in the PsaB and PsaL sequences, potentially resulting in a shifted position of chlorophyll B40 and loss of calcium ion coordination by PsaL. With the construction of a *Synechocystis* mutant containing the *C. aponinum* PsaB sequence, the authors attempted to clarify if the structural changes around chlorophyll B40 could explain the loss of long wavelength chlorophylls and absorbance differences seen in *C. aponinum*. The work carried out by the authors is useful in the understanding of structural and sequence variability in PSI for understanding chlorophyll energy tuning as well as the variability in amino acid sequences throughout cyanobacterial PSI subunits and how they may be adapted for specific light conditions.

The spectral characterization of *C. aponinum* PSI relative to *Synechocystis* provides the crux of the rationale for the work done in the manuscript, however the discussion and interpretation by the authors is limited to primarily the baseline differences between the spectra. As *C. aponinum* is presented as a cyanobacterium isolated via propagation of environmental samples at 3,500 umol photons m-2s^-1^, discussion of the physiological significance of the PSI differences is minimal.

The *C. aponinum* PSI Cryo-EM is well discussed and presented by the authors to highlight the structural features that may be involved in shifted the absorbance spectrum and the loss of far-red wavelength chlorophylls. Calculations of the excitonic interactions based on the solved structure and the shifted position of chlorophyll B40 (relative to the *Synechocystis* position) highlighted the utility of solving the PSI structure rather than relying on sequence differences modeled onto reference structures.

Similarly beneficial to the objective of trying to understand how *C. aponinum* differences impact the function of PSI is the introduction of *C. aponium* PsaB sequence into *Synechocystis* and analysis of the resulting spectrum and far-red chlorophyll presence. As the authors found a large shift in the position of chlorophyll B40 in *C. aponinum* PSI, potentially caused by amino acid changes in the PsaB gene, the introduction and testing in *Synechocystis* is a reasonable first method to understand the impact. The discussion of this chimeric PSI (Red_c) should be addressed better to make clear what is being referenced, the Red_c mutant versus *C. aponinum* itself, as discussion of the differences seems to misstate at times what is actually being compared.

While the manuscript presents data in a clear, concise format, there are some outstanding issues that should be addressed to improve the understanding of the results and significance of the findings.

The selection strategy was using 3,000 umol photons m-2s^-1^ light to select for high light tolerant strains. As the sample was from the Sonoran desert, it would be useful to clarify if/what temperature was used for the selection as well, as the growth temperature of 30C is not particularly high for many cyanobacteria.

The figures detailing the characterization of PSI between different samples could benefit from a clearer delineation between the samples shown. For most samples, the spectra are nearly identical, and the differences, or the fact that two samples are shown is nearly impossible to determine (figures 2C, 4C).

Similarly, it would be useful to discuss the other differences in PSI subunit composition. From the analysis of isolated PSI (figure 2), the authors discuss the migration differences between PsaL. However, there are clear differences between the migration of other subunits as well. Whether the primary sequences are markedly different should be mentioned. The authors also state that the expected number of bands is observed for *C. aponinum* (line 137). It is unclear from the figure if that is the case, as five distinct bands are visible in the *Synechocystis* sample but only four in C. aponinum. If PsaF migrates almost together with PsaD that should be clarified.

Lines 176-177 should be clarify which chlorophylls are discussed relative to the changes in B40 coupling.

The description of the chimeric *Synechocystis* mutant needs considerable clarification. Figure 4B: what is Blue6803? The legend for Figure 2G does not seem to be correct. It should also be clarified that the amino acids present in *C. aponinum* are not mutations (ex: line 253, 399), they are the wild-type sequences and are different relative to Synechocystis. The discussion within the section of the excitonic coupling calculations should also be addressed to clarify the distinction between the *C. aponinum* structure and the Synechocycsist Red_C mutant.

As the PsaB changes around chlorophyll B40 did not explain either the absorbance spectrum changes or the long-wavelength chlorophyll differences, the authors also discuss amino acid differences in the vicinity between the two species. It would be beneficial to do a similar comparison between these positions and amino acid composition between difference cyanobacterial species as was done for the calcium binding residue position. This may also provide more information to discuss these changes rather than a preliminary list of the differences between the two structures.

It is interesting that this mutant did not reproduce either the *C. aponinum* absorbance spectrum or the loss of far-red wavelength chlorophylls, again highlighting the utility of doing both the sequence and structure comparisons. While it would increase the impact of this work to also include mutation comparisons introducing the *C. aponinum* PsaL into Synechocystis, as well as the double *C. aponinum* PsaB/PsaL chimera, that work may not be feasible under current circumstances. However, because the Red_c mutant did not explain the differences between absorbance spectra, a more detailed discussion of how the various changes may be involved in those changes should be carried out.

Based on the presentation in the manuscript, it is unclear why full genome sequencing was performed, as PSI genes could have presumably be sequenced via PCR methods. The authors do themselves a disservice at having done this analysis and essentially not addressing the results. It would be interesting to know if there are any genes within the genome beside those highlighted for PSI that may be of interest in understanding the high-light capacity of this cyanobacterium.

The methods section should be tidied up. Many explanations are unnecessary and likely could be better described via referencing previous works.

In general the manuscript requires further proof-reading throughout to rectify typos, italicization issues, unit labels, etc.

*Reviewer #3:*

In this work, Dobson Z et al., target the PSI complex of a high-light adaptive cyanobacterium *C. aponinum* 0216. The cyanobacterial PSI complexes carry a number of long wavelength chlorophylls (LWCs) in their core antennae. The LWCs were suggested to increase the absorption cross section of PSI, and to serve in photoprotective function. In this work, they suggest that the *C. aponinum* PSI contains less LWC compared with *Synechocystis* PSI, based on their spectroscopic results. They further determine the 2.7 Å resolution cryo-EM structure of PSI complex from the high-light tolerant strain to investigate if it possesses some special structural features. The structure shows that one chlorophyll, B40, shifts its orientation which is likely induced by an inserted loop in PsaB of *C. aponinum* PSI. Therefore, they assume that the altered loop region and chlorophyll B40 are responsible for the spectral property of less LWC in *C. aponinum* PSI. To confirm this assumption, they construct a mutant form of *Synechocystis* strain containing the sequence of the PsaB loop from *C. aponinum*. However, spectra of the mutant PSI indicate that chlorophyll B40 is not responsible for the difference of LWC. Then they analyze the local environment of chlorophylls in *C. aponinum* and *Synechocystis* PSI, and suggest that the local environment of chlorophylls B7-A32-A31 accounts for some of the differences in the absorption spectra.

Overall, the structural data in this work is solid. Generating a mutant strain possessing the different peptide sequence to induce different chlorophyll arrangement based on the structural data is useful in analyzing the spectral property and potential function of individual chlorophyll molecules. It was widely applied in the previous studies to generate a mutant strain possessing different pigment arrangement and to analyze the spectral difference of photosynthetic complexes from the mutant strain, now with the high-resolution structural information, the interpretation of the biochemical and spectral data are more reliable, as the structure provides details regarding the position, orientation and environment of each chlorophyll molecules.

In this work, although the red LWCs or the factors affecting the spectral features of this high-light tolerant cyanobacterial PSI are not identified, they do have some suggestions/ideas worthy of further exploration based on their present results. Therefore the findings about *C. aponinum* PSI are beneficial to the photosynthesis research field, including the artificial photosynthesis research. The conclusions and suggestions put forward by the authors are largely supported by their data, but some aspects of data analysis need to be clarified and amended.

The interpretation of spectral analysis of *C. aponinum* PSI is questionable. They conclude that *C. aponinum* PSI contains less LWC than *Synechocystis* PSI based on the differences of absorption spectra between the two PSIs shown in Figures 2C and 2D. However, the difference is very small, with the amplitude of δ-absorbance lower than 0.04, which seems within the error range, while the absorbance of the PSI complex is around 1.5. On the contrary, the 2-nm red-shift of 77K fluorescence emission of *C. aponinum* PSI is quite evident (Figure 2F). Therefore it is possible that *C. aponinum* PSI contains more, but not less, LWC.

They construct a mutant form of *Synechocystis* strain containing the sequence of the PsaB loop from *C. aponinum*, and find that the 77K fluorescence emission of the mutant PSI is the same with *Synechocystis* PSI. Thus conclude that chlorophyll B40 is not responsible for the different absorption of LWC. Although they confirm the complete and correct replacement of PsaB in the mutant strain, they do not validate that B40 orientates the same with that in *C. aponinum* PSI. Since the loop regions are usually highly flexible and may adopt quite different conformation even with the same primary sequence, it is possible that B40 is indeed responsible for the different absorption of LWC in C. aponinum, but B40 in the mutant does not change its orientation, i.e. adopts the same orientation as that in *Synechocystis* PSI.

*C. aponinum* is a high-light tolerant cyanobacterium, and it does not grow well under low light conditions as shown in Figure 1C. However, in this study, they culture the cells at low light (∼40 μE), which may lead to some changes of the photosynthetic complexes, therefore the structural features of PSI important for *C. aponinum* to survive in the high light environment cannot be identified.This reviewer understands that the difference of absorption spectra between PSI from *C. aponinum* and *Synechocystis* is very small, as only a few chlorophylls are changed. However, if they would like to use the spectral data, they need to prove that the differences of absorption spectra are indeed due to the different spectral property of PSI, but not the standard error. The spectra should be repeated at least three times for each PSI complexes and the differences of the repeated spectra should be shown in the paper.

It's better to determine the structure of the PSI complex from the constructed mutant to show the orientation of chlorophyll B40. If it is the same as that in *C. aponinum* PSI, this result will further strengthen their conclusion. If it is not, then chlorophyll B40 should be further investigated.

It will be more convincing if they purify and determine the structure of the PSI complex from *C. aponinum* cultured under high-light conditions.

Figure 4 please explain Blue6803 and Red_c.

Line 276, B39/B40 -> B19/B40

Line 363, The orientation of the C-terminus -> The orientation of the C-terminus of PsaL

*Reviewer #4:*

In this work the authors present the structure and the biochemical and spectroscopic properties of the photosystem I complex of a high light tolerant cyanobacterium, C. aponimum. They find that this complex has slightly different spectroscopic features compared to PSI from the model cyanobacterium Synechocystis, which grows in low/medium light, and try to relate them to the differences in light growth conditions. Next they proceed to mutate *Synechocystis* PSI introducing some of the structural features present in *C. aponimum* PSI. By comparing the spectroscopic properties of the two complexes, they could identify the absorption property of one specific chlorophyll out of the 96 present in the complex.

The work is interesting and nicely combines structural and spectroscopic data. The conclusions on the second part of the work are sound and show how the protein environment can tune the spectroscopic properties of the pigments. On the other hand the link between the observed PSI features and the high light tolerance of this organism is less convincing.

General comments:

The authors discuss the PSI/PSII ratio of the cells and relate it to the light stress experienced by the two organisms. There are two relevant points

1. The correlation between PSI/PSII ratio and light intensity is less straight forward than indicated in the manuscript. This ratio depends on the species and in plants does not change in different light conditions.

2. The authors estimate the PSI/PSII ratio using the relative intensity of the 680- and 720-nm peaks of the 77-K fluorescence spectra. This quantification is not reliable because it depends on the excited-state lifetimes of PSI and PSII complexes, which might be different in the two organisms. The PSI/PSII ratio should be determined biochemically (i.e. by looking at the protein content) and/or via physiological measurements (electrochromic shift, P700 oxidation rates, etc.). On the other hand, this information is not essential for this manuscript, since the discussion about changes in PSI/PSII ratio is not needed (see the previous point) and PSI is the main focus of the manuscript.

The authors noted – correctly – that the PSI of the high-light growing *C. aponimum* has a lower content of red Chls in comparison to PSI of Synechocystis. However, the correlation of the red Chl content to light stress conditions remains unclear, and the related discussion (page 21) is not very convincing. In general, the authors seem to favor the conclusion that red Chls are detrimental for PSI in high light, which would explain the reduced red Chl content of *C. aponimum* PSI. For instance, the authors hypothesize that *C. aponimum* PSI might carry less red Chls to reduce its trapping time and maximize photoprotection. This claim is not supported by data, since the authors do not measure the trapping time of the two PSI. Moreover, it is known from the analysis of other organisms that the differences in PSI trapping times with different red Chl content are typically small, and that the PSI of cyanobacteria always behaves as a very efficient trap (lifetime < 50 ps) and is therefore well protected independent of the red Chl content. The other hypothesis regarding the IsiA ring (i.e. that the IsiA ring is mainly there to drag excitations away from the red Chls) is also not sound.

The last section in the results is not as strong as the rest of the work and could be removed/reduced. Indeed, the differences in amino acids (mostly aromatic) around Chls B7, A31, A32 do not seem enough to justify the presence/absence of a red Chl cluster. Indeed, red Chls in PSI are usually ascribed to charge-transfer states, which are expected to be influenced more by changes in surrounding charged/polar residues rather than aromatic ones. Also, the mentioned calcium ion seems to be too distant from the Chl cluster to really play an effect in this sense. Finally, the hypothesis of electron donation to a Chl by an aromatic amino acid having a large influence on the spectral properties (discussion, page 20) is not sound from a chemical perspective.

Detailed points:

Lines 34-39. PSI is certainly central in photosynthesis, but I do not see how the authors can conclude it from the fact that the PSI/PSII ratio changes in high light (and please check the recent literature).

Line 55. The discovering of the two photosystems by using different excitations was due to the spectroscopic work of Lou Duysens ( Duysens LNM, Amesz J, Kamp BM (1961) Two photochemical systems in photosynthesis. Nature 190:510-511)

Line 118. A higher carotenoid content relative to chlorophylls is observed in all organisms when grown in high light compared to low light.

Line 120. The ratio between the fluorescence peaks at 77K cannot be used to measure PSI/PSII. The fluorescence signal is much more complex. It can at best give indication that there are changes, but those need to be validated with other techniques.

Line 121-123. The rational for the investigation of PSI does not appear very convincing.

Lines 136-136, Figure 2B. The protein pattern differs between the two PSI. It makes sense that some of the proteins are slightly different, but the authors should not conclude that they could see "the expected bands". The change in mobility of PsaL is explained, but what about PsaF and PsaC?

Line 177. Which two Chls?

Line 233, it should be a different supplementary figure.

Line 277. The decrease is only in the absolute value, because the numbers are negative.

Figure 1. For the dilution experiment, please specify for how long cells were incubated at the given light intensity before taking the picture (C).

Figure 2. the differences in absorption and fluorescence between the PSI of the two strains are very small. Are they reproducible? The normalization is tricky because the authors do not know if the number of pigments is the same in the two PSI.

A number of the references seems to be outdated/incorrect. It is absolutely true that pioneering work of a high quality was done in the 80' and 90', but the introduction of new techniques in the last 30 years has led to the revision of some of these early reports. I would suggest the authors to consult some more recent literature.

Finally, the manuscript needs careful editing.

---

## [Author Response]

Reviewer #1:1) "…had a constant drip of fresh water and exposed to extreme light conditions." How is 1300 uE an extreme light? Are all terrestrial vascular plants now extremophiles?

This raises a good point and we have revised the wording of the text accordingly. That being said, the Sonoran Desert experiences one of the highest light conditions on the planet, experiencing over 300 days of sunlight per year. We theorize that an organism isolated from this type of desert environment could have adapted unique properties. Additionally, the south facing wall (with no shade) provides an optimal environment for studying an organism adapted to experiencing high light. In addition, an even higher light intensity was used during the selection process that ultimately revealed the ability of *C. aponinum* to grow in conditions up to 3000 µmol photons m^-2^s^-1^ to explore the limits of light adaptation.

In our opinion comparing these light intensities to the 200 µmol photons m^-2^s^-1^ used to culture other desert cyanobacteria in (Alwathnani, A. and Jeffrey, R. Cyanobacteria in Soils from a Mojave Desert Ecosystem. 5, 71–89 (2021)) and the 500 µmol photons m^-2^s^-1^ used for 20 minutes to induce high light inducible proteins in *Synechocystis* sp. PCC 6803 (Komenda, J. and Sobotka, R. Biochimica et Biophysica Acta Cyanobacterial high-light-inducible proteins — Protectors of chlorophyll – protein synthesis and assembly. *BBA – Bioenerg.* 1857, 288–295 (2016)) warrants the labeling of *C. aponinum* as an extremophile.

2) In this regard, choosing Synechocystis as a comparison is a bit of a "straw man" and it would be more appropriate to compare to a wider range of species/strains. In other words, wouldn't it be more accurate to say that Synechocystis is adapted to LOW light, rather than C. aponiunum adapted to high? If this is true, then the story may be about selection pressure for low light rather than photoprotectiom.

Our main motivation for using *Synechocystis* is that it is used in laboratories worldwide as the model cyanobacterium leading to its PSI being well characterized. *Synechocystis* can also be genetically modified relatively easily, so we are able to modify its PSI to test how structural variations effect spectroscopic properties. This allows us to validate the significance of the structural differences we observe in *C. aponinum*. Having a well characterized “baseline” to compare is paramount in order to reach any conclusions on the PSI core antenna. Additionally, *Synechocystis* is relatively close to *C. aponinum* on an evolutionary tree. While it may be possible to argue that *Synechocystis* is adapted to low light; “high-light” treatment is usually no more than 500 µmol photons m^-2^s^-1^ for short durations of time (see Muramatsu, M. and Hihara, Y. Acclimation to high-light conditions in cyanobacteria: From gene expression to physiological responses. *J. Plant Res.* 125, 11–39 (2012)).

3) The work shows that C. aponium can grow at higher light than Synechocystis. However, it was not possible to attribute this difference to PSI properties. This would be a pretty high bar to set, of course, but can the authors make the case that the differences they do observe are relevant?

Thank you for this comment. This is an important question. Under high light conditions, PSII is known to become damaged. In *C. aponinum* we show that there is an increased dependence on PSI for cells grown under high light. Even with the contribution of PBS’s, this still means that the balance between the two available photochemical traps changes and PSI is now more dominant. It is true that we cannot yet fully explain how PSI acts to confer high light resistance, but we can show that it is an important conduit of excitation energy at high light. In the revised version of this manuscript, we demonstrate that the red states of PSI are affected by the presence of a ca^2+^ ion located on the luminal side of PSI in the PsaL subunit. This explains the spectroscopic differences between the oligomeric states of PSI, a subject which has been studied for some time, as well as some of the properties of the core PSI complex in the green lineage. We think that identifying precise ways to manipulate the properties of PSI is integral to being able to address its contribution to high light stress and identify the underlying mechanism.

4) The experiments exposed plates to 3000 µmol photons m-2 s^-1^, which is higher than that these cells experience in the real world. What is the argument that 3000 uE is physiologically relevant? Is it not the case that this selects for something not relevant to nature?

3000mE was used during the selection on liquid cultures and not on plates. By exposing cells to extreme environments, we can explore the limits of light adaptation and thereby expose unique properties. This has been done in many genetic screens and has led to the elucidation of highly relevant cellular mechanisms. One example that comes to mind is DNA repair, where cells are often exposed to high levels of UV radiation or DNA damaging agents, still, the repair pathways that were discovered proved to be functional under everyday growth. While 3000mE is not something any cyanobacteria would experience in nature, the fact that *C. aponinum* could survive those conditions was the first indication that photosynthetic organisms can survive high light conditions. *C. aponinum’s* tolerance to high light conditions was also demonstrated on agar plates in direct comparison to *Synechocystis* (Figure 1).

5) The 77K emission spectrum is not a measure of PSI:PSII content, but on the distribution of excitation energy. Further, the interpretation of these spectra is dependent on the specific excitation wavelength. For instance, if cyan light will excite the PBS, which can dump energy differently into PSI and PSII, compared to, e.g. blue light. The emission spectrum is also impacted by cell density and other factors. None of these issues is acknowledged in the text, nor is that excitation wavelength described in the Methods, as is clear from the scant description given in the text: "An Oxford instruments Cryostat was used to cool 26 the sample to 77 K (cells were plunged into liquid nitrogen and measured immersed in liquid nitrogen)." which does not give any indication of the instrument, excitation, cell concentration, etc., though Figure 2 mentions that particular data was collected with excitation at 440 nm. What is the resolution of the spectrometer?

Thank you for this comment. We altered the description in the Results section and improved the description in the method section. We refer to the 77K measurements as a way to approximate the reliance on PSI:PSII, and that is now better reflected in the text. We excited our sample at 440 nm which preferentially excite Chls. While some backflow to PBS may occur, the physiological relevant parameter is the PSI/PSII utilization which ultimately shows more reliance on (or energy distribution to) PSI after growing in high light. We used a Horiba Fluoromax-4 for these measurements with a stated resolution of +/- 0.5 nm. We have indicated our slit settings on both monochromators in the method section for both RT and 77 K measurements.

6) Figure 3. The spectra are interpreted as apparent shifts based on subtracting pairs of spectra normalized to their peaks. This seems to be based on a subjective interpretation and other interpretations are possible. Changing the normalization point could result in an entirely different interpretation. For instance, normalizing to a point on the red side of the peak could give the appearance of an increase in absorbance or emission on the blue edge of the spectrum (rather than a shift). The deconvolution of the absorbance spectra using three Gaussian components as in Figure 2D is likewise based on an assumption or interpretation that there are only these components.

We now normalized the absorbance spectra to the area between 550-775 nm because the structures of each PSI reveal the same number of chlorophylls, making this method a reasonable approximation (this is different from the previous manuscript in which we used the 650-775 nm interval for integration). Normalizing the area between 550-775 normalized the amount of light being absorbed exclusively by chlorophylls. The presence of long wavelength features in the difference spectra persisted using several normalization methods and integration intervals. When computing the deconvolution of the peak, we used the least number of components that adequately fit the data, resulting in 3 total peaks. It’s entirely possible (even probable) that these peaks contain contributions from multiple Chls, we are not assigning these gaussians to specific chlorophylls in our work and our statements that these gaussians represent the contribution of LWC still holds. Methods with higher optical resolution are required to provide finer details on the electronic structure of LWC. Our choice of deconvolution and normalization represents, in our view, the most conservative treatment of our data.

7) The text needs to be more circumspect and/or directly address this as a possibility, since such alternative normalization could indicate a quite different story. For example, the text reads: "The difference spectrum of C. aponinum – Synechocystis shows a strong negative peak at 701 nm revealing that C. aponinum PSI contains less LWC 150 than Synechocystis (Figure 2D)." It suggests this, but there are other interpretations. This issue is more evident further in the manuscript, in Figure 4, a different method of normalization was used, as stated in the following: "The absorbance spectra were normalized to the area between 650 nm to 775 nm and the spectrum of Wild Type Synechocystis was 238 subtracted from Red_c (Figure 4C and 4D)." As argued above, changing the normalization method can give rise to different interpretations, so some real justification must be presented for using a particular approach, and an even stronger argument needs to be made when changing the normalization between samples. None is given.

The indication of different normalization methods were the results of typos that we apologies for, all the previous absorbance data was normalized using the same method to the area between 650 to 775 nm. We completely agree that the same normalization method should be applied to all relevant data and that any changes should be well reasoned. In this version of the manuscript all the presented absorbance spectra were normalized to the area between 550-775 nm to completely account for both Qy and Qx transitions contributions, so comparisons between species are consistent and there are no changes in normalization methods between absorbance samples as we have clarified in the text. Room temperature and 77K emission spectra were normalized to the max wavelength for comparison of peaks ratios and shapes.

8) Figure 4 D compares isolated complexes from two different species and the differences spectra can be interpreted as a series of narrowing and shifting of spectra, but all of which are likely to be dependent on the method of normalization, which is not adequately described. Moever, the method used to make the differences spectra were not consistently applied, e.g. "[t]he absorbance spectra were normalized to the area between 650 nm to 775 nm and the spectrum of Wild Type Synechocystis was subtracted from Red_c (Figure 4C and 4D)." What were the criteria for using the different procedures and how was bias avoided?

This was addressed by our answer above.

9) The shifts in absorbance and fluorescence emission are quite small, especially realizing that the bands tend to sharpen at 77K. At RT, there may be essentially no functionally-relevant differences in the spectra. How can this be addressed?

This is a good point and was addressed in the new version of the manuscript. The addition of a new mutant (*Red_d*) to the updated manuscript shows a difference in LWC at room temperature in both absorption and emission measurements, but not 77K, and therefore we included the room temperature emission of all species/mutants to this version. These differences could be physiological significance, by possibly affecting the rate of chl triplet accumulation or any other damage inducing mechanism known to occur in PSI. Currently, *C. aponinum*, is not genetically amenable and we cannot test these mutations in this organism. On the other hand, transferring them to *Synechocystis* unavoidably leaves additional differences (in PSI and other cellular systems) between these two organisms unaccounted for. Generating a detailed understanding on the mechanisms that tune energy transfer and damage avoidance in PSI can ultimately provide the answer to this question, but the complexity of this system currently prevents a simple explanation, so more work should be done in the future to fully unravel the mechanisms.

10) Can the text be expanded to further discuss the very large apparent difference in PsaF?

To explain this we have run, and included, a new gel that better separate the PsaF and PsaD bands (Figure 2B) and expanded an explanation in the manuscript. In the new gel, size differences can still be seen between some subunits, notably, PsaC, PsaD, and PsaF. We examined the DNA sequences for these genes carefully and found no evidence for additional sequences that can be translated at both the N and C termini. Based on the protein sequence data, the mass of PsaC for *C. aponinum* and *Synechocystis* is 8.82 kD and 8.83 kD, respectively; the mass of PsaD for *C. aponinum* and *Synechocystis* is 16.18 kD and 15.65 kD, respectively; and the mass of PsaF for *C. aponinum* and *Synechocystis* is 18.69 kD and 18.25 kD. These masses do not explain the difference in migration for either species, and the bands we observe on the gel do not correspond precisely to these molecular weights. However, due to the variations of sequences between species, some changes in detergent binding are likely different between these samples. This is known to cause gel shifting, and is common when analyzing membrane proteins via SDS-PAGE and can also account for soluble proteins like PsaC and PsaD. This is well described in (Rath, A., Glibowicka, M., Nadeau, V. G., Chen, G. and Deber, C. M. Detergent binding explains anomalous SDS-PAGE migration of membrane proteins. 106, 1760–1765 (2009)).

11) Cyanobacterium aponinum (C. 78 aponinum). should be both Ital. Is this an official name?

This has been corrected so it is in italics, and it is the official name as deposited in the NCBI database (NCBI:txid2676140). The updated manuscript reflects this in the Results section and the methods section, as well as provided in the data availability statement.

Reviewer #2:While the manuscript presents data in a clear, concise format, there are some outstanding issues that should be addressed to improve the understanding of the results and significance of the findings.The selection strategy was using 3,000 umol photons m-2s^-1^ light to select for high light tolerant strains. As the sample was from the Sonoran desert, it would be useful to clarify if/what temperature was used for the selection as well, as the growth temperature of 30C is not particularly high for many cyanobacteria.

Environmental samples were collected in February 2016. According to the National centers for Environmental Information, the average temperature in February for Tempe, AZ where *C. aponinum* was isolated from is 18.7 C. This information was added to the text to expand on the description of isolation. We used 30 C during selection because that it is used in ours (and others) lab for cyanobacteria cultivation. Over the course of the last five years, we have grown *C. aponinum* at 22 C as well and found that it grows very well at this temperature which is closer to the average environmental value at the time of isolation. Other members of the *aponinum* family were isolated from hot springs and this family of cyanobacteria appear to grow in many different environments.

The figures detailing the characterization of PSI between different samples could benefit from a clearer delineation between the samples shown. For most samples, the spectra are nearly identical, and the differences, or the fact that two samples are shown is nearly impossible to determine (figures 2C, 4C).

This stems from the large number of chlorophylls present in PSI, which means single chlorophyll differences make about 1% of the total Qy intensity. We’ve done our best to visually improve the graphs but this is still a challenge due to the magnitude of the differences. We’ve included difference spectra when appropriate but think that showing the complete spectra is important to accurately report on the extent of the differences and not only their shapes. In emission curves we have included enlarged sections to improve the visibility of the differences.

Similarly, it would be useful to discuss the other differences in PSI subunit composition. From the analysis of isolated PSI (figure 2), the authors discuss the migration differences between PsaL. However, there are clear differences between the migration of other subunits as well. Whether the primary sequences are markedly different should be mentioned. The authors also state that the expected number of bands is observed for C. aponinum (line 137). It is unclear from the figure if that is the case, as five distinct bands are visible in the Synechocystis sample but only four in C. aponinum. If PsaF migrates almost together with PsaD that should be clarified.

To explain this we have run, and included, a new gel that better separate the PsaF and PsaD bands (Figure 2) and expanded an explanation in the manuscript. See our response to reviewer 1, point 10 above.

“Comparing this sample to a known PSI sample from *Synechocystis*, SDS-PAGE shows similar bands for PSI subunits in Figure 2B, with notable shifts in the PsaD, PsaF, PsaL and PsaC subunits.

To investigate the different migration of PSI subunits between *C. aponinum* and *Synechocystis*, *C. aponinum* genomic DNA was isolated and sequenced (NCBI:txid2676140). PSI genes were located and annotated, then compared to *Synechocystis* (Figure 2 —figure supplement 1). The gene for PsaL revealed the difference in migration of is likely due to two substantial differences compared to *Synechocystis*: (1) a 6 amino acid insert located on the stromal side of the membrane between two transmembrane helixes and (2) a markedly different C-terminus (Figure 2 —figure supplement 1) containing an extension of four amino acids. In addition, a 7 amino acid insertion is seen in the PsaB gene of *C. aponinum* compared to *Synechocystis* (Figure 2 —figure supplement 1). Genes for the remaining subunits (PsaD, PsaF, and PsaC) did not reveal size differences that would correspond to these shifts (Figure 2 —figure supplement 1). There were however sequence variations between *C. aponinum* and *Synechocystis* which likely cause gel shifting, a common occurrence when analyzing membrane proteins via SDS-PAGE^60^.”

Lines 176-177 should be clarify which chlorophylls are discussed relative to the changes in B40 coupling.

We have changed this sentence to use chlorophyll numbers in the manuscript text.

The description of the chimeric Synechocystis mutant needs considerable clarification. Figure 4B: what is Blue6803? The legend for Figure 2G does not seem to be correct. It should also be clarified that the amino acids present in C. aponinum are not mutations (ex: line 253, 399), they are the wild-type sequences and are different relative to Synechocystis. The discussion within the section of the excitonic coupling calculations should also be addressed to clarify the distinction between the C. aponinum structure and the Synechocycsist Red_C mutant.

Thank you for pointing these out. Blue6803 was regrettably a typo and has been fixed, it is now referred to as *Red__c_*. We do not have a figure 2G, however we did correct a mistake on the legend of figure 4G. We have clarified that the sequences in *C. aponinum* are natural variations and refer to them as variations/differences in the text. To clarify which models were used in calculations we added the following statement to our method section under “*Excitonic Structure Calculations*:” – “The PDB: *6UZV* model of trimeric PSI was used as *WT Synechocystis* and the *C. aponinum* PSI structure (PDB:6VPV) was used to approximate the Red__c_ structure.”

As the PsaB changes around chlorophyll B40 did not explain either the absorbance spectrum changes or the long-wavelength chlorophyll differences, the authors also discuss amino acid differences in the vicinity between the two species. It would be beneficial to do a similar comparison between these positions and amino acid composition between difference cyanobacterial species as was done for the calcium binding residue position. This may also provide more information to discuss these changes rather than a preliminary list of the differences between the two structures.

We thank the reviewer for this comment. The new results on the *Red_d* mutation now explains some of the differences in the LWC absorption. Additionally, Figure 6 —figure supplement 1 compare the structure of *Synechocystis*, *C. aponinum*, and *Pisum Sativum* (Pea plant), a known eukaryotic PSI structure. Plants have less LWC in its core PSI (Croce, Roberta, Giuseppe Zucchelli, Flavio M. Garlaschi, and Robert C. Jennings. 1998. “A Thermal Broadening Study of the Antenna Chlorophylls in PSI- 200, LHCI, and PSI Core.” Biochemistry.). Further, we have added multiple sequence alignment statistics of the residues shown in Figure 6, to the supplemental figures, investigating the identity of these residues across bacteria and eukaryotic databases (Figure 6 —figure supplements 2-4), demonstrating that these specific residues vary between cyanobacteria and eukaryotic organisms, and that *C. aponinum* contains residues that are more prevalent in eukaryotic organisms than in cyanobacteria. It’s important to note that we don’t want to discount the possibility that water molecules (which we cannot resolve at the current resolution) could be causing the shifts in the spectral characteristics, however the amino acids nearby could also be the underlying reason for a different organization of water molecules.

It is interesting that this mutant did not reproduce either the C. aponinum absorbance spectrum or the loss of far-red wavelength chlorophylls, again highlighting the utility of doing both the sequence and structure comparisons. While it would increase the impact of this work to also include mutation comparisons introducing the C. aponinum PsaL into Synechocystis, as well as the double C. aponinum PsaB/PsaL chimera, that work may not be feasible under current circumstances. However, because the Red_c mutant did not explain the differences between absorbance spectra, a more detailed discussion of how the various changes may be involved in those changes should be carried out.

We included a new mutant, *Red_d*, in this manuscript to explain the consequence of calcium ion binding which partially answers this remark. We agree that generating additional combinations of site mutations is important, however more information is needed before we approach this large task, and we think this should be addressed in a separate work.

Based on the presentation in the manuscript, it is unclear why full genome sequencing was performed, as PSI genes could have presumably be sequenced via PCR methods. The authors do themselves a disservice at having done this analysis and essentially not addressing the results. It would be interesting to know if there are any genes within the genome beside those highlighted for PSI that may be of interest in understanding the high-light capacity of this cyanobacterium.

Because the 16S based tree suggested this is a new species, we sequenced the entire genome to confirm this, and this data is now publicly available for the community to study (NCBI:txid2676140). To enumerate and validate all the sequence changes that contribute to high light resistance requires a separate experimental design, beyond the scope of the current manuscript in our opinion.

The methods section should be tidied up. Many explanations are unnecessary and likely could be better described via referencing previous works.

We have removed some sections (such as phenol chloroform DNA extraction) from the methods section, corrected additional typos and added a unified primers table.

In general the manuscript requires further proof-reading throughout to rectify typos, italicization issues, unit labels, etc.

We have carefully proofread the entire manuscript and made many corrections.

Reviewer #3:The interpretation of spectral analysis of C. aponinum PSI is questionable. They conclude that C. aponinum PSI contains less LWC than Synechocystis PSI based on the differences of absorption spectra between the two PSIs shown in Figures 2C and 2D. However, the difference is very small, with the amplitude of δ-absorbance lower than 0.04, which seems within the error range, while the absorbance of the PSI complex is around 1.5. On the contrary, the 2-nm red-shift of 77K fluorescence emission of C. aponinum PSI is quite evident (Figure 2F). Therefore it is possible that C. aponinum PSI contains more, but not less, LWC.

A 0.04 difference is well above the baseline variation of our spectrophotometer (which is app. 0.0005 OD over the measured wavelength range). To better demonstrate the robustness of this measurement, PSI was isolated from both *C. aponinum* and *Synechocystis* 3 separate times and absorption spectra were taken on 3 separate days and compared to each other. These spectra were then normalized according to our methods section, and the difference spectra of these measurements are shown in Figure 2D, including the standard deviation of the measurements along the entire wavelength range, showing that around the Qy transition (which includes the LWCs) the variation is negligible compared to the size of the difference signal. 77K emission is not a good indication for the total number of LWC, as it is affected by transfer processes that still occur at 77K, therefore this measurement is a better indication of the terminal emitter in PSI. Our hypothesis is that eliminating some of the LWC in *C. aponinum* causes increased energy transfer to a different LWC, which now has a stronger contribution to 77 K emission. This is consistent with the findings that *C. aponinum* contains less LWC but displays a red shifted emission peak at 77K. The principles of this hypothesis are that the contribution of a specific LWC to the 77K emission depends not only on its transition energy but also on its connectivity. This has been shown by many studies, but our recent work on site specific mutations in *Synechocystis* is relevant for this particular case (Khmelnitskiy, A., Toporik, H., Mazor, Y. and Jankowiak, R. On the Red Antenna States of Photosystem I Mutants from Cyanobacteria Synechocystis PCC 6803. *J. Phys. Chem. B* (2020). doi:10.1021/acs.jpcb.0c05201).

They construct a mutant form of Synechocystis strain containing the sequence of the PsaB loop from C. aponinum, and find that the 77K fluorescence emission of the mutant PSI is the same with Synechocystis PSI. Thus conclude that chlorophyll B40 is not responsible for the different absorption of LWC. Although they confirm the complete and correct replacement of PsaB in the mutant strain, they do not validate that B40 orientates the same with that in C. aponinum PSI. Since the loop regions are usually highly flexible and may adopt quite different conformation even with the same primary sequence, it is possible that B40 is indeed responsible for the different absorption of LWC in C. aponinum, but B40 in the mutant does not change its orientation, i.e. adopts the same orientation as that in Synechocystis PSI.

The general statement that loop regions are more flexible than regions with secondary structure is of course correct, but we do not think it applies in this particular case. The first indication for this is the fact that the B40 loop is well resolved in our cryo-EM structure, this means that it adopts a relatively stable conformation in *C. aponinum*. This can be explained by the additional interactions between this loop and the chlorophylls it binds, together with the stacking interactions between the chlorophylls themselves. We have shown previously that chlorophyll binding loops can be transferred between different PSIs, and that the conformation of the loop remain essentially identical to the original structure even in the absence of adjacent subunits (Toporik, H. *et al.* The structure of a red-shifted photosystem I reveals a red site in the core antenna. *Nat. Commun.* (2020). doi:10.1038/s41467-020-18884-w). Further support for our assertion that the B40 loop adopts a similar configuration comes from the high sequence similarity of sequences juxtaposed to this loop, making the immediate environments between the two different PSI’s highly similar (see Figure 3 —figure supplement 3).

C. aponinum is a high-light tolerant cyanobacterium, and it does not grow well under low light conditions as shown in Figure 1C. However, in this study, they culture the cells at low light (∼40 μE), which may lead to some changes of the photosynthetic complexes, therefore the structural features of PSI important for C. aponinum to survive in the high light environment cannot be identified.

We would like to point out that *C. aponinum* grows fine under low light conditions. In the growth study in Figure 1 (and described in the methods and in the updated manuscript’s figure legend), *C. aponinum* was plated at an OD_730_ of 3 and *Synechocystis* was plated at an OD_730_ of 5, and both were diluted in ¼ steps. This was done to best depict the difference in viability between the two cyanobacteria in increasing light. While it would be ideal to grow the cyanobacteria at higher light, technical issues arise, mainly that proteins isolated from stressed cells are not ideal for structural analysis. We also prepared PSI from highlight grown cells and from low light grown cells and did not observe any differences in absorbance or protein content (SDS PAGE), we conclude that PSI itself does not change in response to highlight.

This reviewer understands that the difference of absorption spectra between PSI from C. aponinum and Synechocystis is very small, as only a few chlorophylls are changed. However, if they would like to use the spectral data, they need to prove that the differences of absorption spectra are indeed due to the different spectral property of PSI, but not the standard error. The spectra should be repeated at least three times for each PSI complexes and the differences of the repeated spectra should be shown in the paper.It's better to determine the structure of the PSI complex from the constructed mutant to show the orientation of chlorophyll B40. If it is the same as that in C. aponinum PSI, this result will further strengthen their conclusion. If it is not, then chlorophyll B40 should be further investigated.It will be more convincing if they purify and determine the structure of the PSI complex from C. aponinum cultured under high-light conditions.Figure 4 please explain Blue6803 and Red_c.Line 276, B39/B40 -> B19/B40Line 363, The orientation of the C-terminus -> The orientation of the C-terminus of PsaLReviewer #4:The authors discuss the PSI/PSII ratio of the cells and relate it to the light stress experienced by the two organisms. There are two relevant points1. The correlation between PSI/PSII ratio and light intensity is less straight forward than indicated in the manuscript. This ratio depends on the species and in plants does not change in different light conditions.2. The authors estimate the PSI/PSII ratio using the relative intensity of the 680- and 720-nm peaks of the 77-K fluorescence spectra. This quantification is not reliable because it depends on the excited-state lifetimes of PSI and PSII complexes, which might be different in the two organisms. The PSI/PSII ratio should be determined biochemically (i.e. by looking at the protein content) and/or via physiological measurements (electrochromic shift, P700 oxidation rates, etc.). On the other hand, this information is not essential for this manuscript, since the discussion about changes in PSI/PSII ratio is not needed (see the previous point) and PSI is the main focus of the manuscript.

Reviewer 4 correctly points out that PSI/PSII ratio is complex and adds that these values are a measure of the distribution of excitation energy between photosystems. The distribution of excitation energy between the photosystems is probably a more relevant parameter then the PSI/PSII protein ratio. We have changed the text to better reflect that the 77K measurements is a way to approximate the distribution of excitation energy between PSI and PSII. This demonstrates that under high light, *C. aponinum* directs more excitation energy to PSI, relative to PSII, and this is the opposite of what was reported for *Synechocystis*.

The authors noted – correctly – that the PSI of the high-light growing C. aponimum has a lower content of red Chls in comparison to PSI of Synechocystis. However, the correlation of the red Chl content to light stress conditions remains unclear, and the related discussion (page 21) is not very convincing. In general, the authors seem to favor the conclusion that red Chls are detrimental for PSI in high light, which would explain the reduced red Chl content of C. aponimum PSI. For instance, the authors hypothesize that C. aponimum PSI might carry less red Chls to reduce its trapping time and maximize photoprotection. This claim is not supported by data, since the authors do not measure the trapping time of the two PSI. Moreover, it is known from the analysis of other organisms that the differences in PSI trapping times with different red Chl content are typically small, and that the PSI of cyanobacteria always behaves as a very efficient trap (lifetime < 50 ps) and is therefore well protected independent of the red Chl content. The other hypothesis regarding the IsiA ring (i.e. that the IsiA ring is mainly there to drag excitations away from the red Chls) is also not sound.

In the context of the PSI core antenna, we do not think there is a general all-inclusive idea explaining the physiological role of red chlorophylls. We favor the position that specific LWC (including groups of coupled chlorophylls) contribute to specific functions. A prerequisite for this approach is identifying and being able to manipulate LWC in vivo and in vitro and this work provides an important step towards this goal. We have included a new section in the discussion expanding on this point. We do think that specific red sites are important for interactions with antenna systems (PBS and IsiA) and this is clearly relevant to high light growth. At the same time, we can only propose a general mechanism to account for the contribution of PSI to high light growth in *C. aponinum*.

The last section in the results is not as strong as the rest of the work and could be removed/reduced. Indeed, the differences in amino acids (mostly aromatic) around Chls B7, A31, A32 do not seem enough to justify the presence/absence of a red Chl cluster. Indeed, red Chls in PSI are usually ascribed to charge-transfer states, which are expected to be influenced more by changes in surrounding charged/polar residues rather than aromatic ones. Also, the mentioned calcium ion seems to be too distant from the Chl cluster to really play an effect in this sense. Finally, the hypothesis of electron donation to a Chl by an aromatic amino acid having a large influence on the spectral properties (discussion, page 20) is not sound from a chemical perspective.

The inclusion of *Red_d*, which removes the ca^2+^ ion in question, in this manuscript demonstrates its effect on LWC of *Synechocystis*.

With regards to the distribution of aromatic amino acids, we compared the environment of A31-B7 cluster between *Synechocystis*, *C. aponinum*, diatom, green algae, red algae, and plant PSI, the amino acids discussed are identical between *c. aponinum* and eukaryotic structures and different in *Synechocystis*. Together with the fact that the plant (and green algae) core PSI was shown to contain less LWC than the cyanobacterial PSI (*Synechocystis* and *T. elongatus)* this supports our suggestion that these changes affect LWC is reasonable (see new Figure 6 —figure supplements 2-4). Published structures show that the electrons on the tryptophan indole are 3.1 Å from the chlorophyll ring in *Synechocystis*, which is closer than the ring-ring distance known to effect chlorophyll absorbance. We think this is within the range required to influence the absorbance of a chlorophyll molecule.

Detailed points:Lines 34-39. PSI is certainly central in photosynthesis, but I do not see how the authors can conclude it from the fact that the PSI/PSII ratio changes in high light (and please check the recent literature).

We’ve significantly revised the text to address this issue.

Line 55. The discovering of the two photosystems by using different excitations was due to the spectroscopic work of Lou Duysens ( Duysens LNM, Amesz J, Kamp BM (1961) Two photochemical systems in photosynthesis. Nature 190:510-511)

We included this citation in the current version.

Line 118. A higher carotenoid content relative to chlorophylls is observed in all organisms when grown in high light compared to low light.

We agree and we included a statement regarding this in our text.

Line 120. The ratio between the fluorescence peaks at 77K cannot be used to measure PSI/PSII. The fluorescence signal is much more complex. It can at best give indication that there are changes, but those need to be validated with other techniques.

We have changed the text to better reflect that the 77K measurements is a way to approximate the reliance on PSI:PSII.

Line 121-123. The rational for the investigation of PSI does not appear very convincing.

Our rational is mainly derived from our 77K whole cells measurements. It is very reasonable that other mechanism contributes to the ability of c. aponinum to grow at high light.

Lines 136-136, Figure 2B. The protein pattern differs between the two PSI. It makes sense that some of the proteins are slightly different, but the authors should not conclude that they could see "the expected bands". The change in mobility of PsaL is explained, but what about PsaF and PsaC?

To explain this we have run, and included, a new gel that better separate the PsaF and PsaD bands (Figure 2) and expanded an explanation in the manuscript. See our response to reviewer 1, point 10 above.

Line 177. Which two Chls?Line 233, it should be a different supplementary figure.

This was fixed

Line 277. The decrease is only in the absolute value, because the numbers are negative.

We have changed the text to “decreases in magnitude”.

Figure 1. For the dilution experiment, please specify for how long cells were incubated at the given light intensity before taking the picture (C).

This was added to the description:

“(C) Serial dilutions of *Synechocystis* and *C. aponinum* on BG11 plates. Cells were serially diluted in ¼ steps and incubated at 30°C for 5 days (light intensities > 370 µmol photons m^-2^s^-1^) and 10 days (light intensity = 50 µmol photons m^-2^s^-1^)”

Figure 2. the differences in absorption and fluorescence between the PSI of the two strains are very small. Are they reproducible? The normalization is tricky because the authors do not know if the number of pigments is the same in the two PSI.

This is very reproducible, we modified figure 2D to include the results +- SD. From the structure work, we know that both species of PSI contain the same number of chlorophylls.

A number of the references seems to be outdated/incorrect. It is absolutely true that pioneering work of a high quality was done in the 80' and 90', but the introduction of new techniques in the last 30 years has led to the revision of some of these early reports. I would suggest the authors to consult some more recent literature.

We added citations throughout the manuscript and changed a paragraph in the introduction (lines 42-61) to better acknowledge recent work on PSI photoinhibition:

“… A significant amount of work on photoinhibition has focused on PSII due to its rapid turnover in high-light and the efficient repair mechanisms that evolved to cope with PSII specific photodamage^24,25^. PSI specific damage, however, is irreversible and long lived due to a lack of repair mechanisms, requiring the biosynthesis of new PSI polypeptides^26–30^. Fluctuating light and low temperatures have been attributed to PSI photoinhibition by causing an imbalance in the redox state of PSI donors and acceptors^27,31–34^. …”

Finally, the manuscript needs careful editing.

We carefully edited the manuscript.